



# Simulation of convective moistening of extratropical lower stratosphere using a numerical weather prediction model

Zhipeng Qu[1,2], Yi Huang[1], Paul A. Vaillancourt[2], Jason N. S. Cole[3], Jason A. Milbrandt[2], Man-Kong Yau[1], Kaley Walker[4], Jean de Grandpré[2]

1: Department of Atmospheric and Oceanic Sciences, McGill University, Montreal, QC, Canada

2: Recherche en Prévision Numérique, Environment and Climate Change Canada, Dorval, QC, Canada

3: Canadian Centre for Climate Modelling and Analysis, Environment and Climate Change Canada, Toronto, ON, Canada

4: Department of Physics, University of Toronto, ON, Canada

*Correspondence to*: Zhipeng Qu (zhipeng.qu2@canada.ca) and Yi Huang (yi.huang@mcgill.ca)

**Abstract.** Stratospheric water vapor (SWV) is a climatically important atmospheric constituent due to its impacts on the radiation budget and atmospheric chemical composition. Despite the important role of SWV in the climate system, the processes controlling the distribution and variation of water vapor in the upper troposphere and lower stratosphere (UTLS) are not well understood. In order to better understand the mechanism of transport of water vapor through the
tropopause, this study uses the high resolution Global Environmental Multiscale model of the Environment and Climate Change Canada, to simulate a lower stratosphere moistening event over North America. Satellite remote sensing and aircraft *in situ* observations are used to evaluate the quality of model simulation. The main focus of this study is to evaluate the processes that influence the lower stratosphere water vapor budget, particularly the direct water vapor transport and the moistening due to the ice sublimation. In the high-resolution simulations with horizontal grid-
spacing less than 2.5 km, it is found that the main contribution to lower-stratospheric moistening is the upward transport caused by the breaking of gravity waves. In contrast, for the lower-resolution simulation with horizontal grid-spacing of 10 km, the lower-stratospheric moistening is dominated by the sublimation of ice. In comparison with the aircraft *in situ* observations, the high-resolution simulations predict well the water vapor content in the UTLS, while the lower resolution simulation over-estimates the water vapor content. This overestimation is associated with
the overly abundant ice in the UTLS along with too-high sublimation rate in the lower stratosphere. The results of this study affirm the strong influence of overshooting convection on the lower-stratospheric water vapor and highlight the importance of both dynamics and microphysics in simulating the water vapor distribution in the UTLS region.

## 1. Introduction

Stratospheric water vapor (SWV) strongly influences the Earth radiation budget (IPCC, 2013) and stratospheric
chemistry (e.g., Anderson et al., 2012). Global climate models (GCMs) generally project an increase of SWV during global warming, which may lead to cooling of the stratosphere and further warming of the troposphere and surface (Forster and Shine, 1999, 2002; Solomon et al.,2010) and thus constitutes a potentially important climate feedback mechanism (Dessler et al. 2013; Huang 2013; Huang et al. 2016; Banerjee et al. 2019). Dessler et al. (2013) estimated



the SWV feedback to be +0.3 W/(m² K). Especially emphasized in these previous studies is the importance of SWV in the extratropical lowermost stratosphere, which has the most significant impact on the energy budget at the top of the atmosphere.

Despite its importance, the processes that control the distribution and variation of water vapor in the upper troposphere and lower stratosphere (UTLS) are not well understood. Large discrepancies are found between the "A-Train" satellite observations and the GCMs of the Phase 5 of the Coupled Model Inter-comparison Project (CMIP5) (Jiang et al. 2012). This study shows that the ratio of water vapor content in the GCMs to that from satellite observations can be as large as 2 to 5 in the mid-latitude UTLS region. Such discrepancies cast significant uncertainty in the SWV radiative
feedback simulated by the GCMs (Huang et al. 2016). Global reanalyses also suffer from SWV biases, including the Modern- Era Retrospective Analysis for Research and Applications (MERRA), its newer release MERRA2 and the Interim Reanalysis of the European Centre for Medium-Range Weather Forecasts (ECMWF) (Jiang et al., 2015). One of the motivations of this study is therefore to investigate the possible causes of such overestimation of water vapor in the UTLS in GCM and numerical weather prediction (NWP) models.

The mechanisms controlling the transport of water vapor into the stratosphere are different for the tropical and mid-latitude regions. In the tropical region, water vapor enters the stratosphere primarily through the slow ascent associated with the Brewer-Dobson Circulation (BDC) (Brewer, 1949). The cold temperature of the tropical tropopause layer (TTL) regulates the humidity of the air and therefore is responsible for the moistening of the stratosphere. However,
it remains uncertain how factors such as the temperature in the TTL, strength of the BDC, and the vertical and horizontal mixing are weighted to determine SWV distribution and variation (Fueglistaler et al., 2014). In extra-tropical region, there are several mechanisms that can influence the distribution and variation of the water vapor in the lower stratosphere (Weinstock et al., 2007). Water can be transported to the lowermost extratropical stratosphere by air parcels transported poleward from the TTL and by deep convection. Among these mechanisms, the vertical
transport by mid-latitude convection, although demonstrated to be impactful by studies using *in situ* and remote sensing measurements, remains poorly understood (Poulida et al., 1996; Hegglin et al., 2004; Dessler and Sherwood, 2004; Ray et al., 2004; Hanisco et al., 2007; Weinstock et al., 2007; Smith et al., 2013; Homeyer et al., 2014, 2017; Sun and Hang 2015).

A few studies have attempted to simulate the injection of water into the lower stratosphere using high-resolution NWP models. These studies found that the transport of water vapor into the stratosphere occurs through gravity wave breaking near overshooting tops (e.g., Wang, 2003; Wang et al., 2009, 2011; Homeyer et al., 2017; Dauhut et al. 2018). The overshooting tops form as strong updrafts within convective cells penetrate the stable stratification at the tropopause. They act as obstacles to the lower stratospheric flow and generate gravity waves. In favorable conditions
(Baines, 1995; Sachsperger et al., 2016), the gravity waves break near the cloud overshooting tops, dissipate wave energy through strong turbulence and cause sudden "jumps" of air flow up to more than 2 km height. This upward wind with strong turbulence transports a substantial amont of water vapor and ice into higher altitude in the





stratosphere. The mechanism of gravity wave breaking is well demonstrated, e.g., by Figure 7 in Wang (2003). An associated phenomenon is the so called "jumping *cirrus*" (Fujita 1982), which provide evidences that ice particles are brought into, and potentially hydrate, the lower stratosphere. The mechanism of cross-tropopause transport of humidity associated with gravity wave breaking is generally well simulated using idealized forcing for a short duration

over a limited domain (Wang, 2003; Wang et al., 2009, 2011; Homeyer et al., 2017; Dauhut et al., 2018). In order to evaluate model results against satellite and aircraft measurements it is necessary to develop an experimental framework in which high resolution simulations can be performed over an extended period in which observations are available.

In this study, we use a high-resolution NWP model, Global Environmental Multiscale (GEM), to simulate an observed lower stratospheric moistening event over North America from 26 to 27 Aug 2013 (Smith et al. 2013). The first objective is to evaluate the model capability to successfully simulate the vertical transport of water vapor through mid-latitude tropopause and reproduce the observed increase of lower stratospheric humidity during and after the deep convection event. The second objective is to evaluate, using all available satellite and aircraft measurements, the

simulated water vapor fields at different horizontal resolutions, ranging from low resolution with parameterized deep convection to high resolutions with explicitly simulated convection. The third objective is to compare processes, such as direct water vapor transport versus ice sublimation, that influence the lower stratosphere water vapor budget. In light of the aforementioned lower stratospheric humidity bias in coarse resolution models, we are especially interested to identify possible causes of such biases.

This paper is structured as follows. The next section provides a brief description of the GEM model and the configuration of the simulation experiment, as well as the observation data for comparison and a trajectory model used to link the simulated and observed samples. This is followed by the analysis of the GEM simulation results, with a focus on the lower stratospheric water vapor budget. We then conclude with a summary of the findings and

perspectives for further studies.

## 2. Method

### 2.1 NWP model simulation

The NWP model used in this study is the GEM model of Environment and Climate Change Canada (hereafter ECCC, Côté et al., 1998; Girard et al., 2014). The dynamics of GEM are formulated in terms of the non-hydrostatic primitive

equations with a terrain-following hybrid vertical grid. It can be run as a global model or a limited-area model and is capable of one-way self-nesting. Milbrandt et al. (2016) described the self-nesting configuration with horizontal grid-spacing $\Delta x \geq 2.5$ km; Leroyer et al. (2014) and Bélair et al. (2017) did the same for $\Delta x = 0.25$ km. For the experiments reported here, four self-nested domains are used with areas of 5000x3600, 3000x2000, 1500x1000 and 375x375 km$^2$ which corresponds to horizontal grid-spacings of 10, 2.5, 1 and 0.25 km, respectively. The four nested domains are

shown in Fig. 1. All simulations use 77 vertical levels, with vertical grid-spacing $\Delta z \approx 250$ m in the UTLS region.





For the three high-resolution simulations with 2.5, 1 and 0.25 km horizontal grid-spacings, the double-moment version of the bulk cloud microphysics scheme of Milbrandt and Yau (2005a, b; hereinafter referred to as MY2) is used. This scheme predicts mass and number mixing ratio for each of six hydrometeors including non-precipitating liquid droplets, ice crystals, rain, snow, graupel and hail. Condensation (ice nucleation) is formed only upon reaching grid-scale supersaturation with respect to liquid (ice). For the simulation with 10 km horizontal grid-spacing, the Kain-Fritsch deep convection scheme (Kain and Fritsch, 1990, 1993; hereinafter referred to as KFC) is incurred. The liquid and solid cloud water content from KFC scheme are later passed to the MY2 scheme as hydrometeors of non-precipitating liquid droplet and ice crystal category, respectively.

In addition to the MY2 and KFC schemes, the planetary boundary-layer scheme can also produce implicit clouds, particularly cumulus and stratocumulus (Bélair et al., 2005). It predicts mean liquid and ice water contents as well as cloud fraction. The shallow convection scheme (Bélair et al., 2005) is the third means by which GEM can produce clouds. It predicts mean liquid and ice water contents and cloud fraction for cells that contain shallow cumulus clouds.

The simulation at 10 km grid-spacing is initialized with conditions from the ECCC global atmospheric analysis at 00:00 UTC 24 Aug 2013. It runs for 96 hours until 00:00 UTC 28 Aug 2013. The second nested simulation at 2.5 km grid-spacing runs for the same period of time. The simulations at 1 and 0.25 km grid-spacing are initialized at 12:00 UTC 25 Aug 2013 and run for 24 hours during which the convective event that we focus on in this study developed. Model outputs are saved every 1 minute for the 2.5, 1 and 0.25 km simulations and every 5 minutes for the 10 km simulation.

## 2.2 In situ observation

We use the water vapor measurements from a NASA field campaign, the Studies of Emissions and Atmospheric Composition, Clouds and Climate Coupling by Regional Surveys (SEAC[4]RS, Toon et al., 2016). During this campaign, an ER-2 aircraft provided *in situ* high altitude observations in the UTLS region. These data are used here to verify the model simulated water vapor content at the lower stratosphere. The ER-2 flight on the day of 27 Aug 2013 began from Huston at 16:46 UTC. It performed four descending-ascending movements between ~20 km and ~13 km height crossing the tropopause between 18:00 and 21:00 UTC in an area to the south of the Great Lakes (cyan lines in Fig. 1). The locations where the descending trajectories ended and corresponding times are shown in Fig. 1. The humidity data used here are from the Harvard Lyman-α photo-fragment fluorescence instrument (hereafter named LYA, Hintsa et al., 1999; Weinstock et al., 2009) and are shown in Fig. 2. The corresponding altitude and temperature are also shown in this figure. The measurements with air pressure lower than 115 hPa and water vapor concentration higher than 8 ppmv are marked in all the three panels. These measurements indicate water vapor contents much higher than the standard values (~4 ppmv) in the lower stratosphere.



### 2.3 Back trajectory simulation

In order to include all the different convective events potentially responsible for the moistening of the lower stratosphere captured by the aircraft measurements on 27 Aug 2013, we start the GEM simulations with the largest, low resolution domain (see Fig. 1) and several days earlier. Several mesoscale convective events developed on
different days near the Great Lakes within this domain. To identify the source of water vapor for the aircraft-measured samples, the back trajectories of the air parcels are simulated using the trajectory model, LAGRANTO (Sprenger and Wernli, 2015) and GEM-generated wind fields. Using this technique, we find that the high water vapor anomalies observed by the aircraft in Domain B on 27 Aug 2013 originated from the deep convection events that began at the end of 25 Aug and ended at the beginning of 26 Aug in Domain A (100˚ W to 87.5˚ W, 46˚ N to 50˚ N, ~860x445
km²) as illustrated in Fig. 1 (see more discussions below).

### 3. Model results and analysis

### 3.1 Convective system

We first examine how well GEM simulates the general features of the deep convection events of central interest (within Domain A). Figure 3 shows the brightness temperature for the middle-infrared atmospheric window, which
indicates the cloud top height, synthesized from the GEM simulations at different horizontal resolutions and observed by GOES-13 geostationary satellite (the 11.2 $\mu$m channel, Knapp et al., 2018). The synthetic radiances are calculated for the 10.2-12.2 $\mu$m spectral interval using the Rapid Radiative Transfer Model for GCMs (RRTMG, Mlawer et al, 1997; Iacono et al., 2000, 2008) with GEM-simulated atmospheric and surface properties as inputs. The target convective event simulated by the high-resolution models (2.5, 1 and 0.25 km) begins at around 18:00 UTC 25 Aug.
The convection is initiated a bit later in the 10 km grid-spacing simulation, at around 21:30 UTC. To account for this difference, the synthetic images from model simulations are all taken at 5 hours after the initiation of the convection. The 0.25 km simulation is limited to a small domain due to the limits of computational resources. Its domain is centered where the convection is initiated (95.5 W, 48.5 N). With an eastward movement, parts of the storm system quickly move outside of the simulation domain. We therefore do not include the 0.25 km simulation for the comparison in
Fig. 3.

From Fig. 3, we can identify the location and extent of the convective system from the white-colored areas that signify low cloud top temperatures (high cloud tops). GEM succeeds to predict a strong convective system in the area near the Great Lakes. The locations of the convection are slightly different from one simulation to another. The 10 km
simulation places the convective system west of Lake Superior. The two higher resolution simulations put the same convective system slightly north-west of Lake Superior. The satellite imager shows the storm system over Lake Superior. Another difference is the horizontal extent of the anvil clouds. The two higher resolution simulations generate anvil clouds of very similar forms to the observation. The 10 km simulation however generates clouds that extends in the northeast-southwest direction and covers a noticeably larger area than what is observed by the satellite.
We notice that the 10 km simulation has a larger area with the brightness temperature lower than 210 K (magenta line





highlighted areas) than those in the high-resolution simulations as well as that in the GOES-13 images. These highlighted zones with cold cloud tops represent the intensive convective areas. For all three simulations with different horizontal grid-spacings, the convective areas are all located within Domain A during the 5-hours period after the initiation of convection.

In order to inter-compare the simulations at different resolutions, we perform the evaluation for Domain A which encompasses the convective event of interest in the three simulations at 10, 2.5 and 1 km grid-spacing. The time window for the evaluation is the initial 5 hours of the convection development. During this time, the convection system of interest initiated, developed multiple overshooting tops and moved from the west to the east end of Domain A. At

the end of the evaluation time window, the heights of overshooting tops are observed to generally decrease (not shown), which evidences that the chosen window captures the primary cross-tropopause transport of water by the convective system. Due to its limited domain area, the results of the 0.25 km simulation are not included for Domain A-based inter-comparisons. Instead, the first hour of the convection event in this simulation (before the system begin to move out of the simulation domain illustrated by the green box in Figure 1) is analyzed for comparing some aspects

of the convection (see below).

### 3.2 Overshooting tops and gravity wave breaking

We examine the overshooting tops in the GEM simulation and especially the gravity wave breaking process that was found to primarily account for the water transport into the lower stratosphere in overshooting events. In our simulations, we find both the 1 and 0.25 km simulations generate similar structure of "jumping *cirrus*" to the previous

studies (e.g., Wang et al. 2009, 2011). To illustrate the results, we show in Fig. 4 vertical cross-section in Domain A at 19:49 UTC on 25 Aug from the 1 km simulation. In addition, two movies made from this simulation are included in the Supplementary Materials (Qu, 2019).

Figure 4 shows a few key variables that highlight the impacts of overshooting tops and induced gravity wave breaking.

Two overshooting tops are well identified between the longitudes of 95.94˚ W and 95.57˚ W by the upward extruding isentropic lines in Fig. 4a. The temperature within the overshooting tops are noticeably colder. To the right of the overshooting tops near 95.60˚ W, the region with overturned isentropic lines and convective instability ($d\theta/dz < 0$) is marked by a red circle. As found in the previous studies (e.g., Wang 2003; Wang et al. 2009, 2011), this instability develops in association with the breaking of gravity waves near the overshooting tops. The wave breaking leads to

sudden jump of air flow and transports both ice particles and water vapor upward into higher altitudes in the stratosphere, which is visible from the IWC and water vapor distributions in Fig. 4b and 4c. At the time shown in Fig. 4, the wave breaking region mentioned above is forming a "jumping *cirrus*" patch that is marked by the red arrows. Two other jumping *cirrus,* formed earlier, can be found near -96.10˚ W and -95.75˚ W, as marked by the two magenta arrows. In our simulations, we find that the jumping *cirrus* can extend to 2 to 3 km above the tropopause (~14.5 km,

according to the WMO definition, i.e., the altitude with lapse rates $\Gamma$ decreased to 2 °C km$^{-1}$ or less). The lower stratospheric regions around the jumping *cirrus* are also characterized by high water vapor concentrations (up to 20



ppmv, see Fig. 4c). The water vapor plumes generated are typically 10 x 10 km$^2$ or less in size. However, in some cases, continuously occurring overshooting produces aggregated plumes, which have a size as large as 30 x 30 km$^2$. The continuous development of the overshooting tops, breaking gravity waves and jumping *cirrus* are shown by the movie in the Supplementary Materials (Qu, 2019).

We find in our simulations that the breaking of gravity waves occurs in many ways similar to the breaking of lee waves, which are formed when air flows through a mountain range. On the leeward side of the mountain, when the wave amplitude reaches a critical level, a convectively unstable region develops and consequently leads to wave breaking (Wurtele et al, 1993; Dörnbrack, 1998, Strauss et al, 2005). In the regime of gravity wave breaking, we can

identify a sudden jump of the stratiform flow (Houghton and Kasahara, 1968). In its vicinity, wave energy is dissipated through turbulence which causes a strong mixing. It was found that such wave breaking occurs when the horizontal wind speed perturbation opposes the mean flow and causes stagnation, meeting a prognostic condition (Baines, 1995; Sachsperger et al., 2016):

$$|u'/U| \gtrless 1 \qquad (1)$$

where $U$ is the mean flow speed and $u'$ is the horizontal wind speed perturbation, which can be derived from the vertical wind speed perturbation $w'$ using a 2-dimensional incompressible mass continuity constraint:

$$|\partial u'/\partial x| = |\partial w'/\partial z| \qquad (2)$$

where $x$ is the horizontal distance and $z$ is the vertical height. Larger obstacles will generate larger $w'$ which in turn derives larger $u'$. When $u'$ is large enough to satisfy the condition in eq.1, wave breaking occurs.

Analogies can be drawn to the gravity wave breaking near the overshooting tops. The overshooting tops carry air mass of different horizontal velocity into the lower stratosphere and act to block the pre-existing horizontal flow there (westerlies, speeds ranging from 5 to 25 m/s at different altitudes). The obstructed stratified flow in the lower stratosphere is forced to pass around the overshooting tops, creating a similar situation to air flow passing a mountain

range and inducing gravity waves.

Fig. 5 shows several key variables for the same cross-section shown in Fig. 4 but 4 minutes earlier (19:42 UTC) when the condition of gravity wave breaking (eq. 1) is satisfied. At this moment before the formation of "jumping *cirrus*", the cloud overshooting top near -95.6˚ W is falling after reaching its maximal altitude with a speed of ~-10 m s$^{-1}$ (Fig.

5c, dark blue area between the altitude of 15 and 16 km). This downward movement will eventually bring the majority of overshot ice and vapor back to the equilibrium level below the tropopause. The falling of the overshooting top generates a strong vertical wind speed gradient (right hand side of eq. 2) just above the overshooting top near the altitude of 16 km. Using eq. 2, we calculated the corresponding horizontal wind speed perturbation (u') shown in Fig. 5d. An area with strong u' with maximal value of ~6 m s$^{-1}$ can be found above the falling cloud overshooting top as

pointed by the red arrow. This value is of the same range as the background horizontal wind speed in the west-east



direction at the altitude of 16 km (~5 m s$^{-1}$ on Fig. 5e). The condition in eq. 1 is then satisfied which leads to the stagnation of air flow and the breaking of gravity wave. The breaking of gravity wave further formed the jumping *cirrus* shown in Fig. 4 (red arrow). In this process, a significant amount of ice and water vapor is transported irreversibly into the lower stratosphere instead of falling back to the equilibrium level.

We emphasize the "irreversibility" of the vertical upward transport during the gravity wave breaking event. In case of a non-breaking gravity wave, the ascending air will later descend after reaching the wave ridge. In this case, the upward transport is "reversible". In addition, there is weaker turbulent mixing to bring up the moister air below because the wave energy is less transferred to turbulence but is propagated away. In the case of gravity wave breaking,

a sudden jump of air flow occurs. The wave energy is dissipated through turbulence in the vicinity of the jump which enhances the mixing and the transport of water vapor and ice from upper troposphere to lower stratosphere.

The occurrence of gravity wave breaking depends on the intensity of the overshooting strength. As shown in eq. 2, the magnitude of the horizontal speed perturbation is linked to the vertical wind speed perturbation, which in this case is related to the overshooting strength. This is in agreement with the finding of Dauhut et al. (2018) that stronger

overshooting tops favor wave breaking and thus facilitate more vertical water transport.

We find in our simulations that the overshooting tops and wave breaking are frequently observed in the 0.25 km and 1 km simulations, with the breaking waves and jumping *cirrus* of typical horizontal sizes of 2 to 3 km. These phenomena are visible in the 2.5 km simulation, although with less frequency and intensity, and are not found in the

10 km simulation because the grid size cannot resolve the process.

### 3.3 Humidity field

We further examine the water vapor fields simulated by GEM at different horizontal grid-spacings. Fig. 6a and 6b show the mean vertical profiles of water vapor volume mixing ratio and temperature within the afore-defined Domain A and 5-hour time window. All the simulations show noticeable enhancement of moisture above 15 km, which bends

the vertical trends of the humidity profiles and produces 'bumps' (elevated water vapor contents) above the tropopause (indicated by the circles in Fig. 6, hereinafter the tropopause is defined by the altitude where the mean lapse rate $\Gamma$ within Domain A and 5-hour time window decreased to 2˚C km$^{-1}$ or less). Surprisingly, the low resolution (10 km) simulation predicts the highest water vapor content in large part of the UTLS. This is interesting because, as described above, the overshooting tops and the gravity wave breaking processes are only resolved in the higher resolution

simulations, which causes vertical water transport and explains the moistened lower stratosphere in these simulations. The moister lower stratosphere in the 10 km simulation (compared to the 1 and 2.5 km simulations) warrants further investigation. In this subsection, we first use satellite remote sensing and aircraft *in situ* observations to validate the vertical humidity profiles, to find out which simulation better approximates the reality, and then, in the following subsection, we diagnose the causes of the identified model biases.




It is challenging to observe the humidity at the levels near tropopause. Nadir-view satellite remote sensing instruments, such as the Atmospheric Infrared Sounder (AIRS) onboard NASA Aqua satellite, usually cannot accurately measure the low water vapor concentrations in the lower stratosphere (e.g., Divakarla et al., 2006), although attempts have been made to improve the retrieval under special circumstance (e.g., Feng and Huang 2018). Limb-view sounders

such as the Microwave Limb Sounder (MLS) on board Aura satellite have higher sensitivity and provide measurements of water vapor content in the UTLS, although biases have also been noted in these datasets. For instance, an underestimation of water vapor with mean bias up to -25% and biases for individual case up to -85% between 100 and 300 hPa were reported for the MLS retrieval (Read et al., 2007; Vëmel et al., 2007, Hegglin et al., 2013; Livesey et al., 2018). It is important to bear in mind the uncertainty in the satellite data when comparing the

simulations with the observations.

Figure 6c, 6d show the comparisons between the GEM simulations and MLS observations. Because of the scarcity of the collocated satellite data and also the afore-mentioned mismatch in time and location of the simulated convective system, we conduct the comparison with respect to area-averages rather than individual samples. The MLS

measurements used here include five MLS footprints located between [38˚N, 45˚N] and [95˚W 93˚W], taken on 26 Aug 2013 around 19:00 UTC (about 15 hours after the dissipation of the convection system). The GEM results used here are the mean values of the simulated humidity profiles within the 100x100 km regions centered on the MLS footprints. The comparison here suggests that the higher-resolution simulation better approximates the satellite observations while the 10 km simulation is found to overestimate the water vapor content in the UTLS region.

Although the concentration values are generally lower in the MLS observations, this may be subject to the aforementioned negative bias in the MLS data. We next compare GEM to aircraft *in situ* measurements to further investigate the humidity biases.

High accuracy hygrometers onboard high-altitude aircrafts provide benchmark water vapor measurements, although

the temporal and spatial coverage of the aircraft data are limited. On 27 Aug 2013, the ER-2 aircraft deployed in the SEAC[4]RS field campaign obtained UTLS water vapor measurements located to the southwest of the Great Lakes as shown in Fig. 1. Using back-trajectory calculations we find that the measured air samples in Domain B are in the downwind direction of our studied convective system in Domain A. For the back-trajectory calculations we use the wind field simulated by GEM at 2.5 km grid-spacing. It is used to trace air parcels at 56 locations on 6 vertical levels

between 15 and 17.5 km and 500 m intervals. Figure 7 shows the back trajectories of 16 selected air parcels starting at the altitude of 15.5 km and at the time of 19:40 UTC, 27 Aug around which time the aircraft measurements were taken. Through the back trajectories, we find that most of the air parcels in Domain B previously passed through Domain A where the convective system analyzed above developed. The high moisture content samples located in the southwest corner of Domain B are especially found to be moistened by the overshooting convection. Figure 8

illustrates the evolution of a few properties of one of these convectively moistened air parcels located in the southwest corner of Domain B. The back trajectories for the other 5 initial altitudes give similar conclusions and are not shown. The results here establish a connection between Domain A and Domain B. It is particularly evidenced that the ER-2





aircraft measured UTLS air samples characterize the moistening effects of the overshooting convection that occurred earlier (and analyzed above) in Domain A.

Given the variability of water vapor in Domain B as shown in Fig. 7 and possible errors in the location and time of GEM-simulated water vapor features, it would not be meaningful to compare the aircraft measurements with GEM simulations at the exactly matched locations and times. Instead, we compare the mean vertical profiles averaged for Domain B between the aircraft observations and GEM simulations in Fig. 6e, 6f. We note that only the 10 and 2.5 km simulations cover Domain B where aircraft data are available (see Fig. 1). We observe a slight temperature bias from both model simulations of ~2 K above the tropopause level at 15.5 km (Fig. 6f). For water vapor content, Fig. 6e shows that the 2.5 km model well predicts the aircraft measurements, which indicates noticeable moistening around the altitude of 16.5 km. Although it also captures this moistening feature, the 10 km simulation generally overestimates the water vapor contents in the UTLS region. This is consistent with the comparison made in Domain A against satellite measurements (see Fig. 6c and discussions above). This noticeable moist bias in the 10 km simulation warrants an investigation of its cause. As the convective parameterization is turned on at this resolution, both resolved vertical air motions and parameterized vertical transport (via the KFC scheme) potentially account for the convective moistening in GEM. This is similar to the simulations of GCMs, which also generally overestimate the UTLS humidity. In the next subsection, we diagnose how an overly moist UTLS occurred in the coarse resolution simulation.

### 3.4 Budget analysis

We diagnose the water vapor transported across the tropopause into the lower stratosphere in the GEM simulations. The water budget is calculated for the rectangular box surrounded by a given lower boundary (e.g. tropopause) and model top (~30 km) as well as the four lateral facets of Domain A. Using the wind, water vapor and tendency fields generated by GEM, we calculate the contributions to the change in total water vapor in the stratospheric box due to vertical advection of water vapor, as well as the sublimation of ice.

First, we use the Reynolds decomposition (eq. 3-5) to diagnose the vertical transport of water vapor for the two high resolution simulations (1 and 2.5 km grid-spacings). For the 10 km grid-spacing simulation, the vertical advection is composed of two parts: the grid-scale advection which is solved explicitly and the parameterized sub-grid scale transport (tendency on water vapor due to KFC) which makes this case not suitable for Reynolds decomposition.

$$w(x,y,t) = \overline{w(x,y)} + w(x,y,t)' \tag{3}$$

$$q(x,y,t) = \overline{q(x,y)} + q(x,y,t)' \tag{4}$$

$$\sum_{nt=1}^{nt=N}\sum_{ns=1}^{ns=M} \delta t \delta s w(x,y,t)q(x,y,t) =$$
$$N\delta t \sum_{ns=1}^{ns=M} \overline{\delta s w(x,y)q(x,y)} + \sum_{n=1}^{n=N}\sum_{ns=1}^{ns=M} \delta t \delta s w(x,y,t)'q(x,y,t)' + \tag{5}$$
$$\sum_{n=1}^{n=N}\sum_{ns=1}^{ns=M} \delta t \overline{\delta s w(x,y)}q(x,y,t)' + \sum_{n=1}^{n=N}\sum_{ns=1}^{ns=M} \delta t \delta s w(x,y,t)'\overline{q(x,y)}$$





We decompose the vertical wind speed $w$ and humidity $q$ into the time averaged terms and fluctuation terms as shown in Eq. (3) and (4), where $x$ and $y$ represent the coordinates of longitude and latitude and $t$ represents the time. The integrated product of $w$ and $q$ for the evaluation domain and time is shown in the Eq. (5), where $N$ is the total number of time steps of the simulation during the evaluation window, M is the total number of horizontal grid boxes in the

Domain A, $\delta t$ is the length of each time step, $\delta s$ is the horizontal surface of a given model grid box.

Applying Eq. (5) at the tropopause level in Domain A (mean $\Gamma$<2˚C km⁻¹), we obtain a first-order approximation of the vertical transport of water vapor through the tropopause. Among the four terms on the right side of the Eq. (5), the last two terms are negligible. The first term on the right side of the equation measures the transport of water vapor

through tropopause by the mean updraft (or downdraft). The second term on the right side of the equation includes transport by "eddies" generated by wave breaking. Through the decomposition for the 1 km simulation, we find that the first term represents 39% of the total vertical transport and the second term represents 59%, which highlights the important role of wave breaking. With the decrease of model's horizontal resolution, the weight of the eddy term decreases to 29% for the 2.5 km simulation. This suggests that the importance of wave breaking in direct vertical

transport of water vapor decrease with the model resolution.

The comparison between the high-resolution simulations and the 10 km simulation is less straightforward because their tropopause heights are different (Fig. 6a and 6b). We therefore calculated the water vapor change due to the vertical advection and ice sublimation with different altitude levels as the lower boundary from 14 km to 16 km. These

results are shown in Fig. 9.

The vertical advection simulated by the high-resolution models is relatively constant from 14.5 km to 16 km altitude with positive values (upward transport, Fig. 9a). This upward advection is linked to the gravity wave breaking (see discussion in Subsection 3.2) which makes the stratospheric air flow 'jump' by about 2 km upward and transports

humidity into the stratosphere. The vertical advection in the 1 km simulation is generally stronger than that of 2.5 km simulation due to the more important role of wave breaking. This is in agreement with the results from Reynolds decomposition.

It is not a surprise that the higher resolution NWP models tend to produce stronger vertical transports across the

tropopause because, as shown in Subsection 3.1, the transport is closely related to the strength of overshooting and the breaking of the gravity waves. Similar to what was found by Weisman et al. (1997), we find in our GEM simulations that the simulated maximal vertical wind speed is inversely proportional to the horizontal grid-spacing of the NWP model. The stronger overshooting wind speed in the higher resolution simulations leads to favorable conditions for gravity wave breaking (see the discussions in Subsection 3.1) and thus more vertical transport.


The total vertical advection from 10 km simulation can be decomposed into two parts: the grid scale explicit advection (blue dashed line in Fig. 9a) and the sub-grid scale advection by KFC (blue dotted line in Fig. 9a). The grid scale





vertical advection of 10 km simulation is positive below the altitude of 14.3 km. It turns to negative values from 14.4 km. One of the reasons for these negative values may be the lack of representation of gravity wave breaking. Another reason is possibly the large-scale circulation induced by the convection. In the convective area, the air transported to the level above 14 km is relatively dry and cold, whereas the descending areas surrounding the convective zone are moister due to the sublimation of the ice. The sub-grid scale advection is strongly negative in lower altitude. In KFC, this downward transport comes from the effect of compensating subsidence outside of the convective updrafts. This strong downward transport gradually reduces to zero near the tropopause. Using tropopauses (as highlighted by circles in Fig. 9a) as lower boundaries for budget analysis, we notice that the higher resolution simulations tend to have more upward vertical advection of water vapor with $3.2 \times 10^8$, $1.1 \times 10^8$ and $-2.4 \times 10^8$ kg for 1, 2.5 and 10 km simulation respectively.

We find large discrepancies in the contribution of ice sublimation throughout the UTLS region between the high-resolution simulations (1 and 2.5 km) and the 10 km simulations. The value of ice sublimation used here includes all the ice-phase categories. For the two high resolution simulations, the contribution of ice sublimation reaches their maximal positive values near the altitude of 14.5 km. Toward higher altitude, the contribution of ice sublimation decreases gradually to a small value. On the other hand, the contribution of ice sublimation from 10 km simulation is large and always positive. Using tropopause as lower boundaries, we find that the higher resolution simulations tend to have less contribution by ice sublimation with $0.4 \times 10^8$, $1.6 \times 10^8$, $3.5 \times 10^8$ kg for 1, 2.5 and 10 km simulations respectively (highlighted by circles in Fig. 9b).

An additional comparison including the 0.25 km simulation for a smaller domain (see Fig. 1) and shorter period (1 hour) corroborates the above finding that simulation tends to have a larger contribution from advection and less contribution from sublimation as the resolution increases (see Supplementary Fig. SI.1, Qu, 2019).

The combined effect of vertical advection and ice sublimation is shown on Fig. 9c. Results indicate a maximum production of water vapor for the 10 km simulation and slightly larger values for the 1 km experiment compare with the 2.5 km version. These results explain the differences seen in the average humidity profiles on Fig. 6a which shows indeed that the 10 km simulation has the moistest atmosphere below 16 km.

The mechanism of transporting water vapor into the lower stratosphere is different between the high- and low-resolution simulations. For the high-resolution simulations, the transport is always positive and is contributed by both vertical advection and ice sublimation. Advection plays a major role in the vertical transport of water vapor and contributes to 89% of total transport at the tropopause level for the 1 km simulation and makes up to 40% for the 2.5 km simulation. The ice sublimation has also a non-negligible contribution for the 1 km simulation (11%) and a more important role for 2.5 km simulation (60%). The transport of water vapor in the 10 km simulation is different. The primary source of moistening of the UTLS region above the altitude of 14 km is the ice sublimation. Throughout the region the amount of water vapor produced by ice sublimation is large and can exceed by more than one order of



magnitude the contribution from the two high resolution simulations. Vertical advection dose not contribute to the moistening of the UTLS region but transports a significant part of water vapor back to lower altitudes. Overall, the contribution from both processes generate a strong moistening of the UTLS region for the 10 km simulation. If we use the high-resolution model as the reference, these results suggest that the 10 km model with deep convection

parameterization dose not well represent the physical mechanisms of transport of water vapor in the UTLS region and the fact that the amount of water vapor from ice sublimation is over-estimated.

It is, however, interesting that compared to the high-resolution simulations, the sublimation induced lower stratospheric moistening is stronger in the lower resolution simulations (10 km grid-spacing), as shown by Fig. 9b.

We find that this higher sublimation rate may be contributed for the following reasons: First, more abundant ice particles in the lower stratosphere in the 10 km simulation, as shown by Figure 10. The cause of this higher mean ice water content may be due to the lack of the parameterization of downward transport to bring the ice within the cloud overshooting tops back into the upper troposphere in the KFC deep convection scheme. In this scheme, the ice transported into the lower stratosphere by the parameterized updrafts will be distributed uniformly into the $10 \times 10$ km$^2$

model grid box. The ice will then be passed into the MY2 microphysical scheme in the hydrometeor "ice" category. Part of these 'ice' particles will eventually be transformed through aggregation or diffusional growth into other larger hydrometeors such as "snow". But all these solid hydrometeors can only fall back into the upper troposphere through gravitational sedimentation. In the reality, and as evident from the high-resolution simulations here, the majority of ice transported into lower stratosphere within the cloud overshooting tops will be brought back to troposphere when

the overshooting tops fall back to the equilibrium level. This suggests that the simple entrainment-detrainment model in KFC might not well represent the complexity of the different mechanisms near the tropopause level.

A separate test run with 10 km model grid-spacing without the KFC scheme has been performed and shows that the ice water content at the UTLS region above the altitude of 14 km is very close to the 2.5 km simulation (Fig. 10a,

purple line). These results suggest that the over-estimation of ice is partly due to the use of the deep convection parameterization. With the reduction of ice in the test run, we find that the mean ice sublimation tendency is largely reduced (blue line and purple line in Fig. 10b). However, with similar amount of ice in the UTLS, the 10 km simulation without KFC still shows a higher ice sublimation rate above the altitude of 14.5 km compared with the two high resolution simulations.

This second reason leading to stronger moistening of the lower stratosphere in the coarse (10 km) resolution simulation is due to the sublimation efficiency of ice. We find that the ice transported into lower stratosphere is largely sublimated into water vapor in the 10 km simulation and much less for the higher resolution simulations. The amount of ice transported across the tropopause at the 1 and 2.5 km simulations are similar: 2.3 and 2.6 $\times 10^9$ kg respectively out of

which only 2% and 6% are sublimated. In contrast, in the 10 km simulation, the vertically transported ice across the similar altitude (~14.5 km) is $3.9 \times 10^9$ kg and 21% is sublimated. If evaluated at the tropopause determined from the 10 km simulation (~15.5 km, higher than the tropopause in the higher resolution simulations), 75% of the $4.8 \times 10^8$ kg





ice is sublimated in the 10 km simulation. In summary, the coarse resolution simulation strongly over-estimates the fraction of ice that is sublimated in the lower stratosphere comparing with higher resolution experiments.

What leads to the drastically different ice sublimation processes in the coarse-resolution simulation? We find that one important factor influencing ice sublimation efficiency in the lower stratosphere is the spatial distribution of ice. Fig. 11 shows the ice sublimation rates, IWC distributions and a few related fields, from the GEM simulations at different resolutions, at the level of ~15.36 km in Domain A at 23:25 UTC 25 Aug 2013. From the panel Fig. 11a, we can identify that the 10 km simulation is populated with many pixels with high sublimation tendency (in yellow) corresponding to the edges of convective cloud areas. These areas are in-between the oversaturated air (relative humidity over ice ~1.08) within the convection clouds and the surrounding dry mid-latitude lower stratospheric air (Fig. 11d, g). The ice sublimation rates in the higher resolution simulations are very different. In these higher resolution cases, the areas loaded with ice are much smaller than that in the 10 km simulation and are of much higher spatial heterogeneity. The majority of the ice is concentrated in the very limited areas (Fig. 11e, f) with low temperature (Fig. 11k, l) and high relative humidity (Fig. 11h, i), corresponding to the locations of the overshooting tops. That is, the majority of ice is *"trapped"* in the overshooting tops. As shown by Fig. 12, this trapping effect is also evidenced by the distributions of ice with respect to temperature (panel a) and relative humidity (panel b), respectively, calculated by summing up the mass of ice in the grid boxes whose temperature or relatively humidity values fall within each specific interval and then dividing it by the total mass of ice in the whole domain. We find that the majority of ice of the two high resolution simulations are trapped in cold temperatures between 195 and 201 K and high relative humidity inside the overshooting tops. The horizontal extent of the areas with high ice water content is thus small. The trapped ice therefore has less contact surface with the surrounding dryer stratospheric air. This factor significantly limits the ice sublimation rate in the higher resolution simulations. In contrast, in the 10 km simulation, the ice is not trapped in the cold overshooting tops but distributed over larger areas with the warmer temperatures (> 201 K). This leads to significant larger contact area with dry air and higher sublimation.

## 4. Conclusions and discussions

In this study we use the GEM model of ECCC to reproduce a mid-latitude lower stratospheric moistening event over North America near the Great Lakes during 25-26 Aug 2013. Simulations are conducted with a set of nested domains at increasing resolutions from 10 km to 0.25 km grid-spacing. Satellite remote sensing data from MLS as well as aircraft *in situ* observations from the SEAC[4]RS campaign are used to evaluate model simulations, complemented with trajectory simulations to associate those observations with model forecasts over specific regions. Comparisons conducted here suggest that while the higher resolution simulations well approximate the observed water vapor fields in the UTLS region after the deep convective events, the coarse resolution simulation simulates a substantially moister UTLS.





Intercomparison of simulations using different horizontal resolutions, we find that the high-resolution simulations (with grid-spacing dx ⩽ 1 km) can properly resolve the key dynamical features of the overshooting convection, including the overshooting tops, the gravity wave breaking process and the visible jumping *cirrus* phenomenon. The overshooting convection may significantly elevate the water contents (both vapor and ice) up to 1-2 km above the tropopause. The size of the high-concentration water vapor plumes is typically less than 10 km although they can aggregate to sizes greater than 30 km. Coarse resolution simulations (dx ⩾ 10 km) cannot resolve these features, although a moister UTLS region results from the parameterized deep convection and associated water transport.

A lower stratospheric water budget has been performed to quantify the contributions by different processes. It shows that vertical advection of water vapor is one main contributor to the lower stratospheric moistening in the overshooting events. In the high-resolution simulations (0.25 and 1 km) where the gravity wave breaking process is well simulated, eddies resulting from wave breaking are found to mainly account for the direct vertical transport of water vapor into the lower stratosphere. This transport mechanism is largely dependent on the strength of overshooting (updraft speed), with higher resolution simulations generating stronger updrafts which enhance overall the transport of water vapor into the stratosphere.

Another important source of water vapor in the lower stratosphere is ice sublimation. The comparisons conducted in this study show that the 10 km simulation has considerably higher ice sublimation rate. One of the possible reasons is that the KFC convective scheme that is turned on in this simulation bring more ice into the UTLS region which enhances the production of water vapor through the ice sublimation process. The cause of this overproduction of ice is likely associated with the lack of downward transport of ice that is observed after the cloud overshooting tops reach their maximal height in the lower stratosphere. The simple entrainment-detrainment model used in KFC scheme may not well represent the complex processes near and above tropopause during the convective event. One solution is to add an element in the KFC scheme to take into account this downward transport above the tropopause. Another solution is to increase the ice particle size in the UTLS so that the ice can sediment faster and hence reduce the ice water content at these levels. Another possible reason why the 10 km model has higher sublimation rate is the high ice sublimation efficiency. This high efficiency is due to the different distribution of ice water contents comparing to the those of high-resolution models. This results from the inability to resolve overshooting tops by the coarse grid boxes ($10 \times 10$ km$^2$) and the failure to represent the trapping of ice by the cold air within the overshooting tops. One possible solution to address this problem is to add a parameterization to reduce the ice sublimation rate in the lower stratosphere or to condition it on the sub-grid scale temperature or relative humidity variability. The moist bias identified in the coarse resolution simulation of GEM here is reminiscent of the moist bias in many GCMs. The ideas for remedying this issue stimulated by the diagnoses here warrant further investigations.





**Author contribution**

Y. H., P. A. V., J. N. S. C., M-K. Y. and K. A. W. conceptualized the research goals and aims. Z. Q. and Y. H. designed the experiments and Z. Q. carried them out. Z. Q. developed the model code, performed the simulations and prepared the manuscript with contributions from all co-authors.

**Competing interests**

The authors declare that they have no conflict of interest.

**Acknowledgment**

The authors thank Katja Winger and Sylvie Leroyer for their help with running the high-resolution GEM model, and Yuwei Wang for his help with running the RRTMG model. This research is supported by a grant from the Atmospheric
Science Data Analysis program of the Canadian Space Agency (Grant number: 16SUASURDC).

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



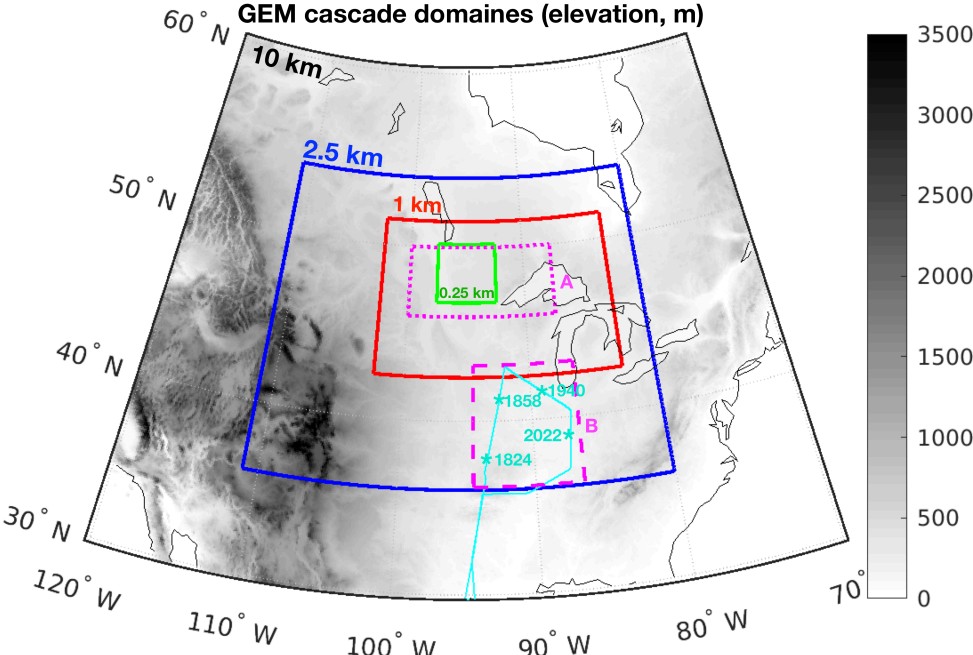

Figure 1: GEM cascade domains. Thick solid lines in black, blue, red, and green represent simulation domains at 10, 2.5, 1 and 0.25 km horizontal grid-spacings, respectively. Thin cyan line represents the ER-2 aircraft flight path on the day of 27 Aug 2013. Magenta dotted line represents evaluation Domain A which covers the major convective events of this study. Magenta dashed line represents Domain B for comparison between aircraft observations and model simulations. The four cyan stars show the locations and times (UTC) of the lowest location of the descending-ascending trajectories of the aircraft.

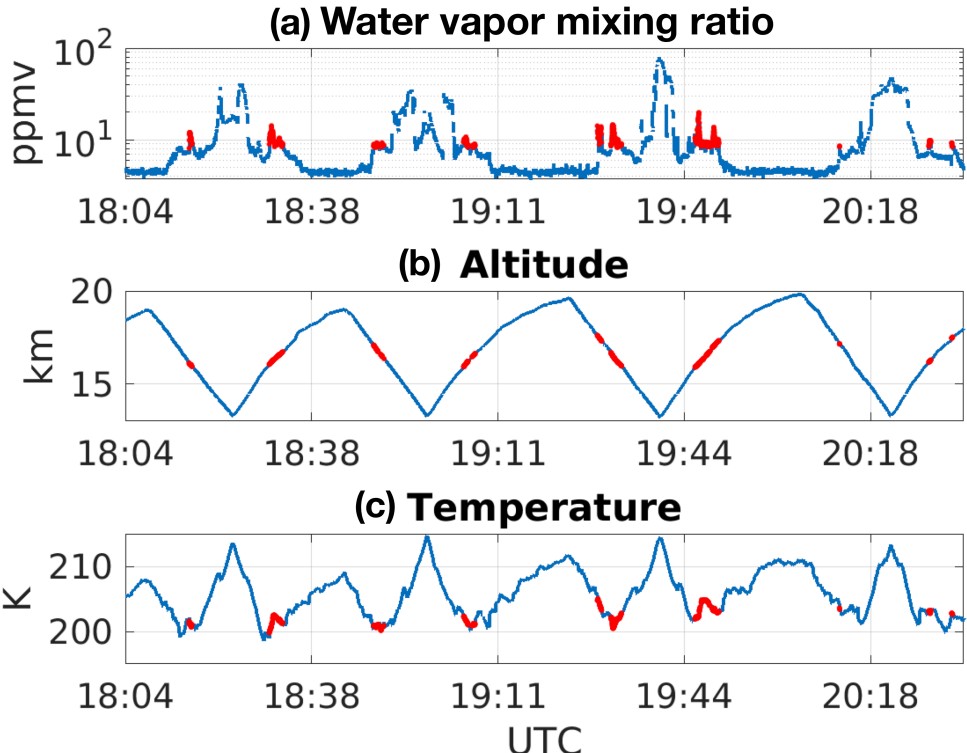

**Figure 2:** ER-2 aircraft observation of a) water vapor mixing ratio in ppmv in logarithmic scale, b) altitude in km and c) air temperature in K. The red dots highlighted the measurements with pressure lower than 115 hPa (tropopause) and water vapor volume mixing ratio greater than 8 ppmv.

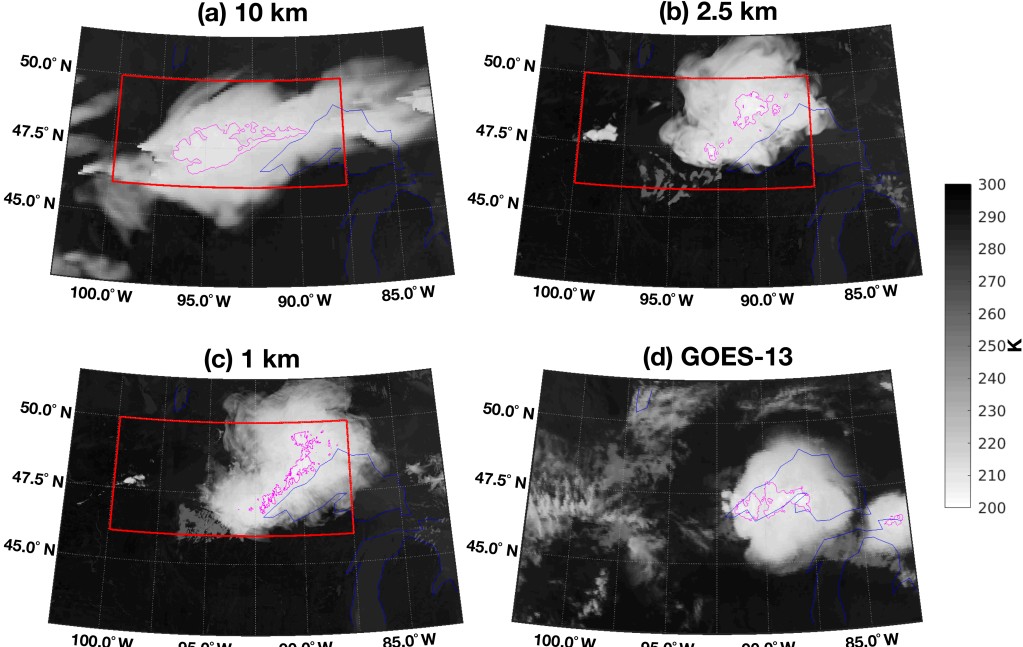

**Figure 3: GEM-simulated deep convective clouds compared to satellite observation. Brightness temperatures are simulated using the RRTMG radiative transfer model, from GEM simulations at three resolutions with 10 km (02:30 UTC 26 Aug), 2.5 km (23:00 UTC 25 Aug) and 1 km (23:00 UTC 25 Aug) grid-spacing, and compared to the brightness temperature of 11.2 μm channel of GOES-13 satellite (06:00 UTC 26 Aug). The red rectangles mark the Domain A. The magenta lines highlight the area with brightness temperature lower than 210 K.**



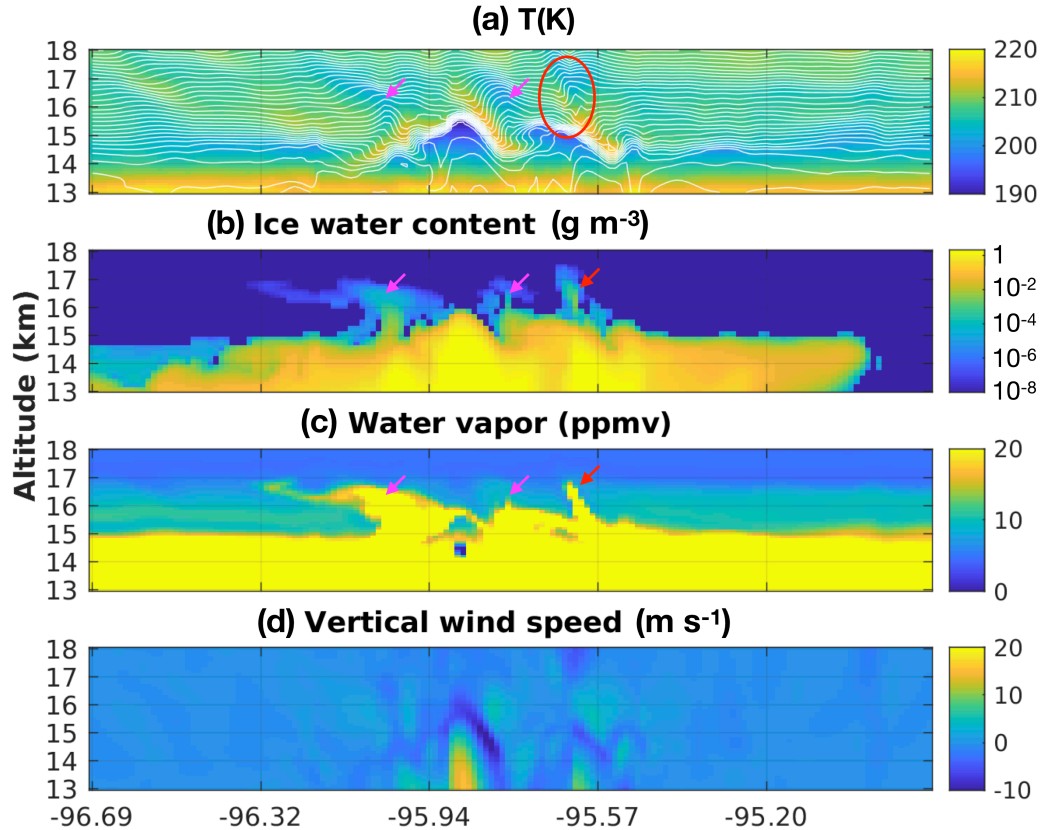

**Figure 4: GEM-simulated overshooting convection. The results illustrated here are taken from the 1 km simulation at 19:46 UTC 25 Aug 2013: a) temperature (colour) and potential temperature (thin white lines); b) ice water content, in logarithmic scale; c) water vapor mixing ratio in ppmv; d) vertical wind speed (m s⁻¹).**





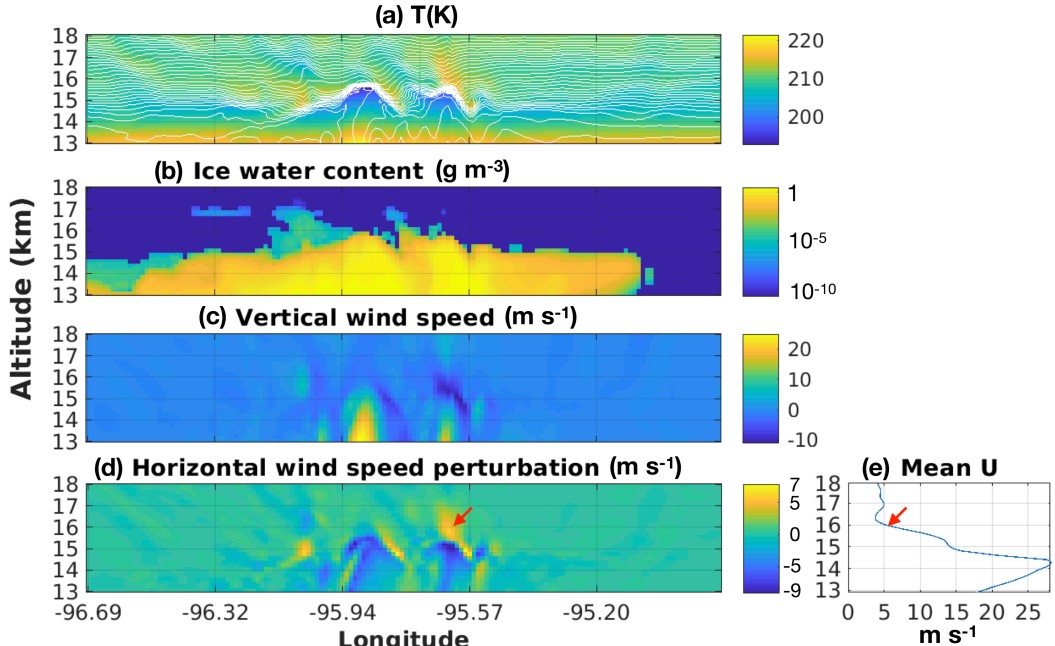

**Figure 5: cross-section as shown in Fig. 4 but for 19:42 UTC 25 Aug 2013: a) temperature (colour) and potential temperature (thin white lines); b) ice water content, in logarithmic scale; c) vertical wind speed (m s$^{-1}$); d) horizontal wind speed perturbation based on eq. 2 (m s$^{-1}$); e) the mean horizontal wind speed on west-east direction.**



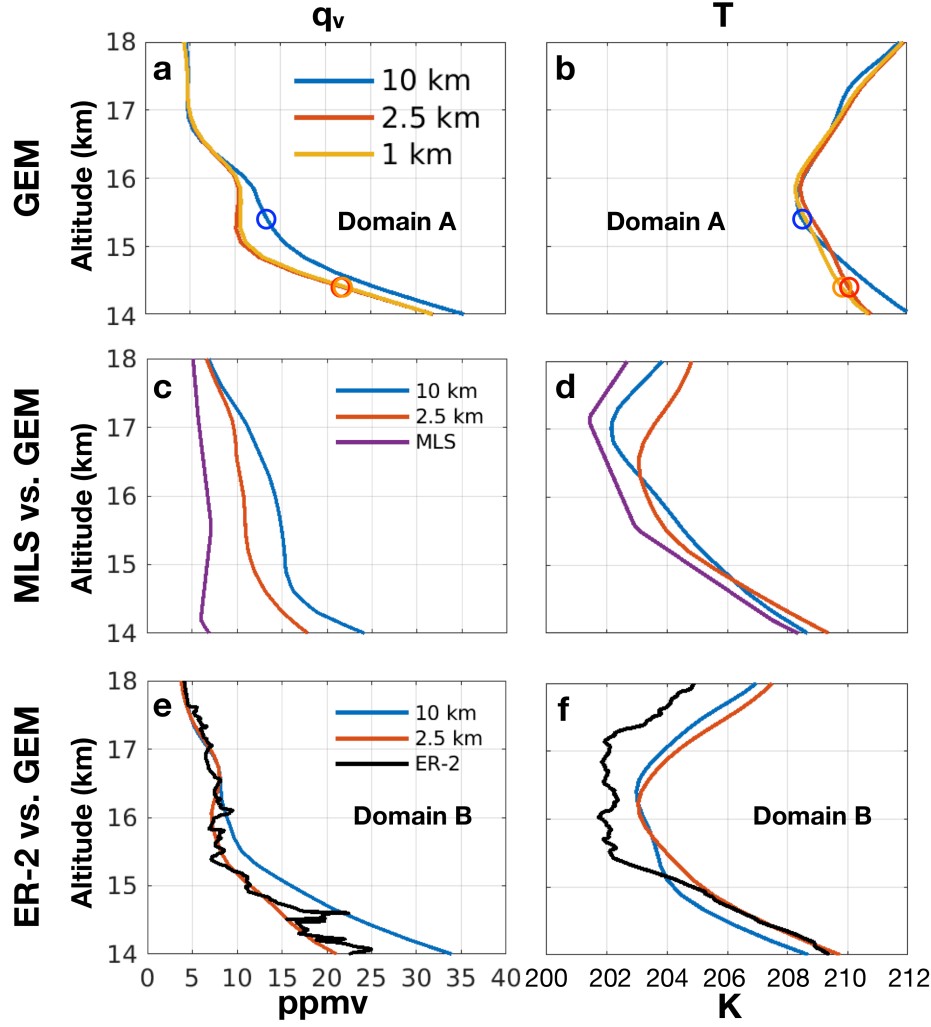

**Figure 6: a), b) the mean profiles of water vapor volume mixing ratio ($q_v$) and temperature (T) for Domain A during the 5-hour period. The circles indicate the positions of the tropopause (mean $\Gamma$<2°C km$^{-1}$); c), d) mean profiles ($q_v$ and T) of GEM 2.5 km, 10 km simulation (100x100 km areas centred on 5 MLS footprints) and MLS data; e), f) the vertical profiles ($q_v$ and T) within Domain B for GEM 10, 2.5 km simulations and for ER-2 aircraft *in situ* observations.**

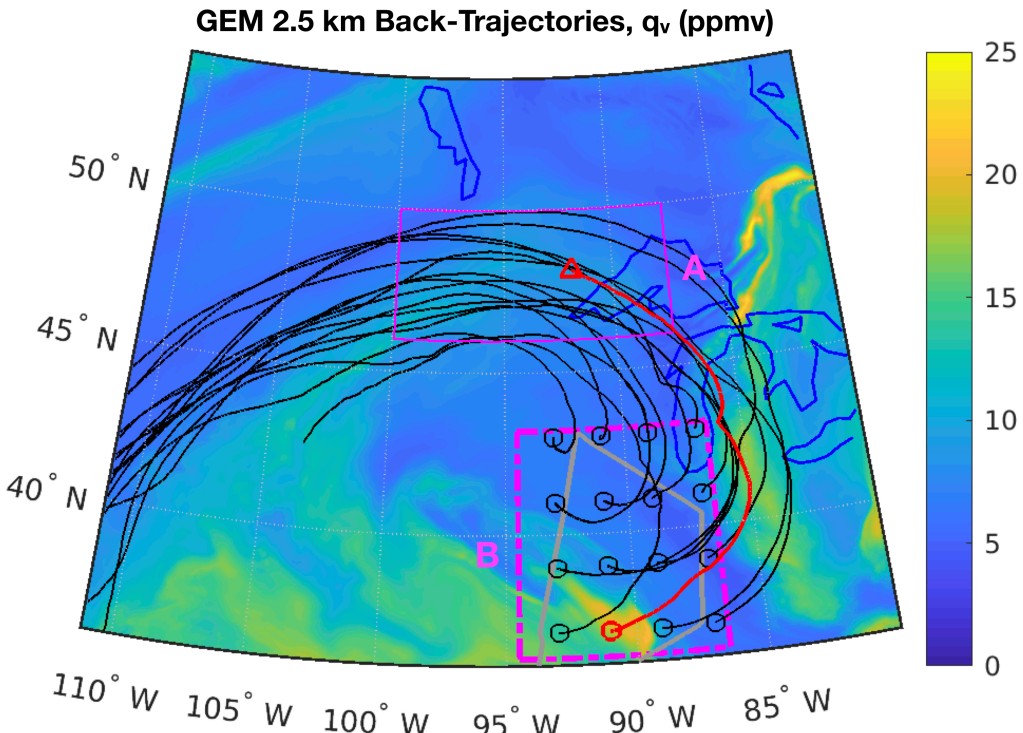

**Figure 7: Back trajectories of air parcels. All the trajectories, in black lines, are initialized in Domain B at 15.5 km altitude and 19:40 UTC 27 Aug. The circles indicate the initial locations of each back trajectory. The gray line here illustrates the ER-2 aircraft flight path (in clockwise direction). The background image shows the water vapor content in ppmv at this level from the 2.5 km simulation. The red line highlights the back trajectory of one air parcel from its initial location in Domain B to its location in Domain A at around 23:00 UTC 25 Aug 2013 when the overshooting convection occurred. The evolution of the properties of this air parcel are shown in Fig. 8.**



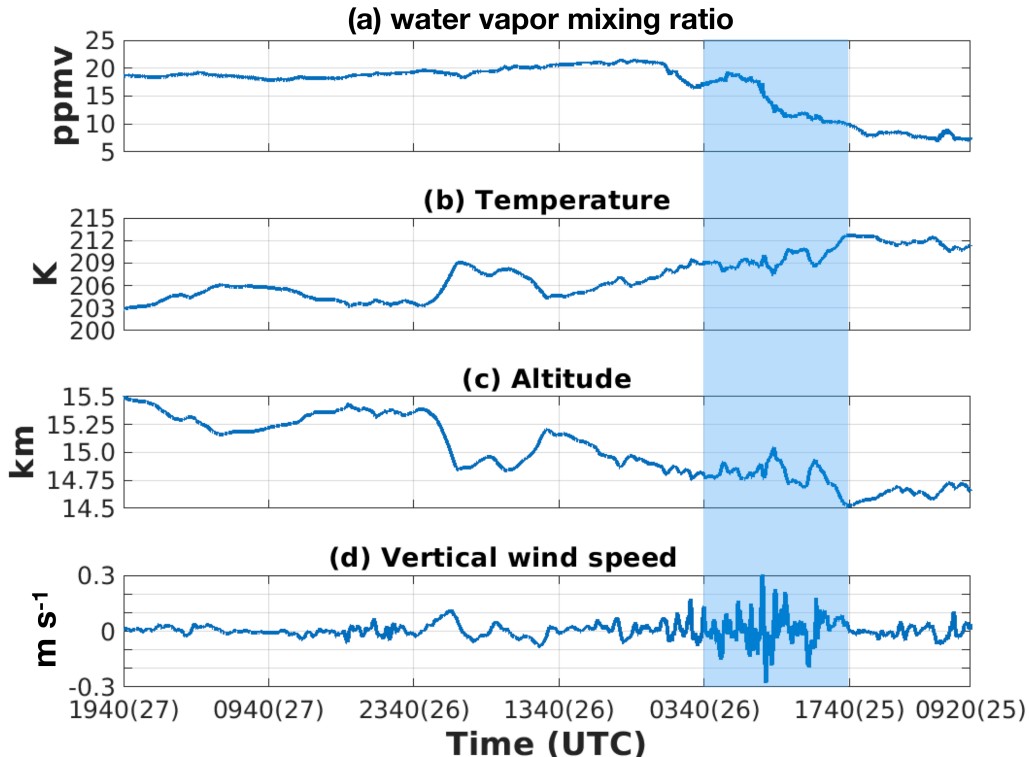

**Figure 8: The changes of the properties of a tracked air parcel along its back trajectory. The beginning time of the back tracing is 19:40 UTC 27 Aug. Highlighted in the rectangular shaded area is the encountering of the air parcel with the overshooting convective region in Domain A at around 22:40 UTC 25 Aug which is evidenced by fast rising water vapor concentration, decrease in temperature, rise in altitude and sudden changes in vertical wind speeds.**



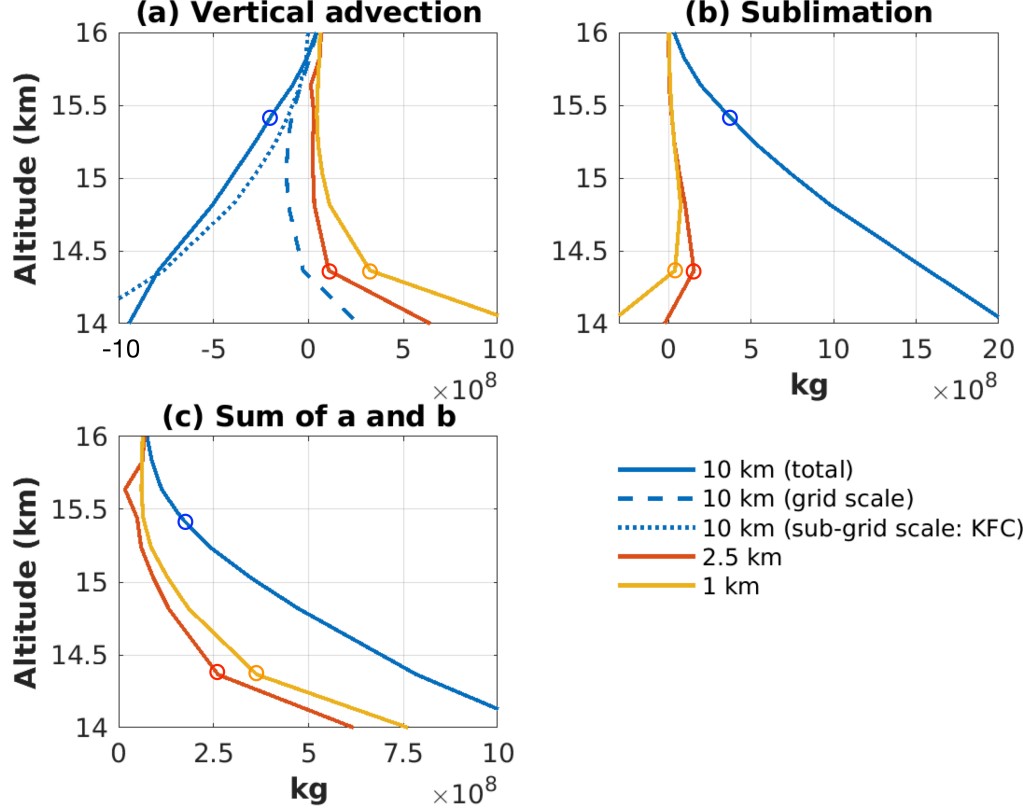

**Figure 9: Change of water vapor in Domain A during the 5-hour evaluation period with different altitude levels as the lower boundary. The circles represent the height of tropopause (mean $\Gamma < 2°C$ km$^{-1}$).**



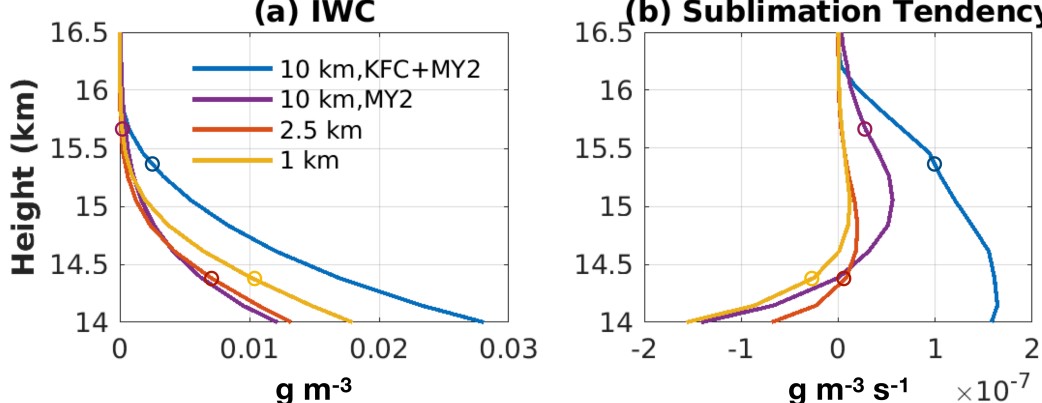

**Figure 10: mean profiles between 14 km and 16.5 km within Domain A and evaluation time window (5 hours): a) ice water content; b) ice sublimation tendency. The circles indicate the position of tropopause (mean $\Gamma < 2°C$ km⁻¹).**

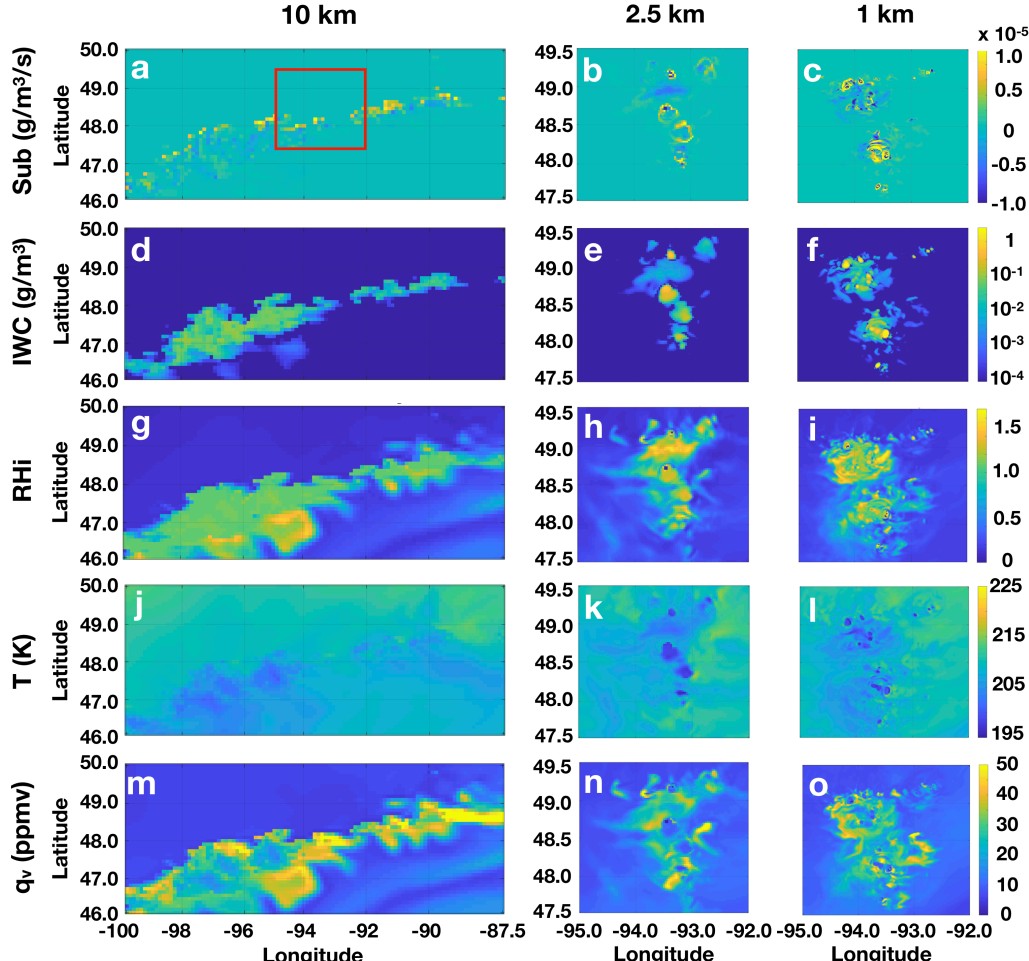

**Figure 11: Distribution of a few variables in Domain A during overshooting. The results are taken from one instance: 23:25 UTC, 25 Aug. 2013. and one vertical level of ~15.36 km (above the tropopause). Shown in the three columns are the simulations at 10 km, 2.5 km and 1 km horizontal grid-spacing, respectively. The area shown for the 2.5 and 1 km simulations corresponds to the red rectangle in the first image. Short names for each row are used for notation, Sub: sublimation; IWC: ice water content, RH: relative humidity with regard to ice, T: temperature; $q_v$: water vapor volume mixing ratio.**





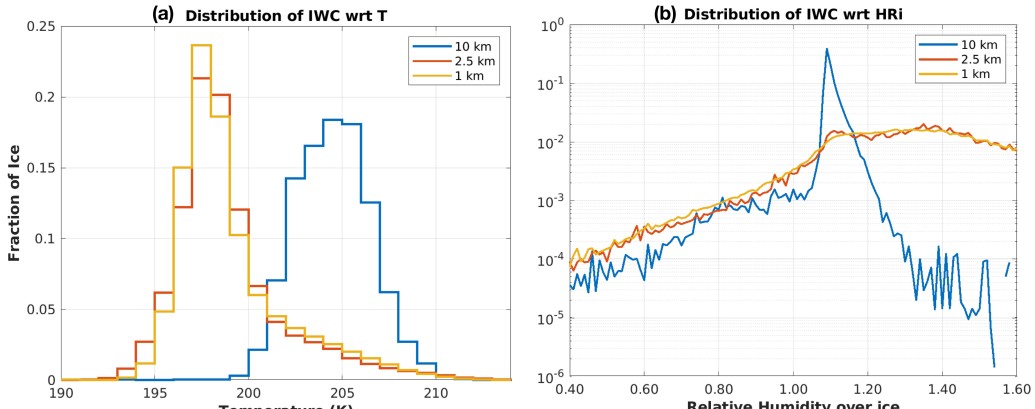

**Figure 12: Distribution of ice with respect to a) temperature and b) relative humidity. The results are based on the variables in Domain A at as shown in Figure 11. The temperature and relative humidity intervals are 1 K and 1%, respectively. The mass fraction value at each temperature or relative humidity interval is calculated**

5 **by summing up the mass of ice in the grid boxes whose temperature or relatively humidity values fall within the specific interval and then divide by the total mass of ice in the whole domain.**