# Peer review of "Simulation of convective moistening of extratropical lower stratosphere using a numerical weather prediction model"

_Atmospheric Chemistry and Physics, 2019_

## Referee Comment (RC1) · Anonymous Referee #2 · 10 Oct 2019

**1   Content**

This manuscript describes model simulation of an overshooting convective event occurred in 2013 over North America. The simulation are performed with different setups and resolutions. The authors found an moistening of the lower stratosphere by the overshooting convection. They further concluded from the detailed model simulation that the main processes for moistening are breaking gravity waves induced by the convection and sublimation of ice crystals. In addition, the simulations are compared to aircraft in-situ measurements and satellite observations.

[Figure]

**2  Overall impression and rating**

The overall impression of the manuscript is good. The manuscript is well structured and the text is easy to understand. The simulations and process interpretation are an important contribution to the community. For these reasons, I recommend publication in ACP after minor revisions.

**3  Specific comments/questions:**

- Page 1, line 4: Maybe add here Riese et al. 2012. They showed also how small change in water vapor due to mixing processes change the radiative budget of the UTLS.

- Page 2, line 24: Isentropic transport of water vapor due to planetary wave activity is also an important transport mechanism for transporting tropical tropospheric air into the lower extra-tropical stratosphere (see e.g. McIntyre and Palmer, 1983; Waugh, 1996; Homeyer and Bowman, 2012 ). I recommend to include this mechanism also in the manuscript to complete all transport pathways.

- Page 2, line 33: I recommend to add the paper of Lee et al., 2019. They performed also high resolution model simulation of an overshooting event in the Asian monsoon region and showed how the moistening occurred and how the hydrated air was transported in the lower stratosphere.

- Page 4, line 34: In summer times the standard value in the extra-tropical lower stratosphere is more 5 ppmv (see Zahn et al. 2014 Figure 5). I would recommend to change the text from 4 to 5 ppmv.

- Page 6, line 21: Here you state the time of Fig. 4 to be at 19:49. In the figure caption it is stated 19:46. Please correct one of these times.

- Where there any cloud instrumentation aboard the ER-2 for measuring cloud number concentration or IWC? If yes, did you check if there were still ice crystals present at flight altitude in the domain B. That would be interesting to see, because than the ice crystals would have been transported over a longer distance in the stratosphere. This transport is shown by the Lee et al. 2019 and it would be interesting to see, if it occurred also in your case.

- Page 9, lines 5-10: Maybe it is worth a mention that the averaging kernels of limb sounders like MLS smear out the strong vertical gradient in water vapor at the tropopause (Hegglin et al.,2013).

- Page 9: The comparison between GEM and MLS is not really done in a balanced way. The difference is partly larger (> 100%) than the baises which are reported in the literature. This strong differences can be hardly explained by just mentioned the possible bias of MLS water vapor. It could also be a result of the model simulation. For example, warmer temperatures in comparison to MLS could lead to slower ice crystal growth and thus less dehydration and thus higher gas-phase water, which could be transported into the lower stratosphere.

Can you please comment on the following questions and suggestions:

- Which Version of MLS data are you using (this would be also nice to mention in the text)?

- Did you applied the averaging kernels of MLS onto the GEM profile ? Otherwise a fair comparison is barely possible. Why not using the exact location of MLS and applying the averaging kernels?

- Why are the profiles of water vapor (panel a/c/e) and also temperature (panel b/d/f) so different ? They should both represent air masses moving from domain a to domain b as shown by the trajectories. It

seems that the situation is strongly variable. Can please discuss in the text about the standard deviation of the mean profiles to get a better feeling on the variability.

- For better understanding of the deviation between GEM and MLS, I would also recommend to add the location of the MLS profile also into the map in Figure 7.

- Figure 8: You show some parameters of an individual trajectory in this figure. The water vapor amount with ~20 ppmv is quite high and the temperatures are cold between 203-206K, which could create conditions with supersaturation wrt. ice. and therefore additional ice formation. Do you see any signs of ice formation in the lower stratosphere along these trajectories? This could be important, because it could partly dehydrate the previously hydrated air masses.

- Page 10-11: For me it is not clear, which part of the equation 5 account for ice sublimation/transport ? Because in the equation only q, which stands for water vapor, is considered. Can you please better explain how you estimated the change due to advection and ice sublimation as stated in lines 16-20.

- Page 12-14: Where does the sublimation occurs in the hight resolution models ? Is it directly in the overshoot or are the ice crystals first mixed into the lower stratosphere by wave breaking and small scale mixing and than sublimate ? Which brings me to a further question, if ice crystals are transported along the trajectories in the lower stratosphere ? Perhaps, you can add this information also into the manuscript.

- Page 13/14: What does a negative sublimation tendency mean ? I would guess it is additional ice formation or particle growth. Or is it both ?

- Page 14: I agree with your conclusion that ice sublimation occurs less pronounced in the high resolution model because of existence of ice crystals mostly

in the overshooting core. In the relative humidity distribution (Figure 12 b) I would expect higher fraction of IWC in the sub-saturated region (Rhi < 1) for the low resolution compared to the high resolution, if there is a much higher sublimation rate. Do you have an explanation for the agreement of IWC distribution in sub-saturation for all three model setups ?

**4  Technical comments/suggestions:**

- Page 8, line 8: Citation should be Vömel et al.

- Page 14, line 9: It is more common to use the term supersaturated instead of oversaturated.

- Figure 2: Can you please include the date of the ER-2 measurements into the figure caption.

- Figure 4: The comprehensibility of the vertical wind speed in panel d would be better, if you choose a color scale centered with the color white at the value of 0 and with positive/negative values in two different colors (e.g. red and blue).

- Figure 5: Same suggestion above for panel c and d.

- Figure 12, caption: "relative humidity" instead of "relatively humidity"

**5  References:**

- Homeyer, C. R., and K. P. Bowman (2012), Rossby wave breaking and transport between the tropics and extratropics above the subtropical jet, J. Atmos. Sci., 70, 607-626.

[Figure]

- Lee, K.-O., Dauhut, T., Chaboureau, J.-P., Khaykin, S., Krämer, M., and Rolf, C.: Convective hydration in the tropical tropopause layer during the StratoClim aircraft campaign: pathway of an observed hydration patch, Atmos. Chem. Phys., 19, 11803–11820, https://doi.org/10.5194/acp-19-11803-2019, 2019.

- McIntyre, M. E., and T. N. Palmer (1983), Breaking planetary waves in the stratosphere, Nature, 305, 593-600.

- Riese, M., F. Ploeger, A. Rap, B. Vogel, P. Konopka, M. Dameris, and P. Forster, Impact of uncertainties in atmospheric mixing on simulated UTLS composition and related radiative effects, J. Geophys. Res., 117, D16305, doi: 10.1029/2012JD017751, 2012.

- Waugh, D. W. (1996), Seasonal variation of isentropic transport out of the tropical stratosphere, J. Geophys. Res., 101, 4007-4023

- Zahn, A., Christner, E., van Velthoven, P. F. J., Rauthe-Schoch, A., and Brenninkmeijer, C. A. M.: Processes controlling water vapor in the upper troposphere/lowermost stratosphere: An analysis of 8years of monthly measurements by the IAGOS-CARIBIC observatory, J. Geophys. Res.-Atmos., 119, 11505–11525, https://doi.org/10.1002/2014JD021687, 2014.
* * *

---

## Referee Comment (RC2) · Eric Jensen (Referee) · 4 Nov 2019

**Review of "Simulation of convective moistening of extratropical lower stratosphere using a numerical weather prediction model" by Z. Qu et al.**

This manuscript uses cloud-resolving model simulations with different horizontal resolutions and different treatments of deep convection to investigate the physical processes leading to hydration of the lower stratosphere by overshooting deep convection. The manuscript is generally clear and well written. The results help clarify the roles of advection, wave breaking, and sublimation in hydration of the lower stratosphere. I have a number of minor comments I would like the authors to consider before

publication.

**Suggestions for authors**

1. It would be helpful if the authors would provide some information about the simulated microphysical properties of the convective tops extending into the stratosphere. Specifically, quantitative information about the simulated ice concentrations and size distributions would be helpful. I realize the simulations use bulk parameterizations, but the two-moment scheme should provide ice concentrations and some measure of the width of the assumed size distribution. Realistic treatment of ice microphysics is important because the simulated convective hydration depends in part on the ice crystal size-dependent sedimentation and sublimation of ice in the lower stratosphere.

2. Page 1, line 30: The authors cite Anderson et al. (2012) here for the influence of water vapor on stratospheric chemistry. I believe earlier references such as *Solomon et al.* (1986) would be more appropriate.

3. Page 2, lines 26–20: I would recommend citing *Smith et al.* (2017) here.

3. Page 3, lines 17–19: It is my understanding that global models generally do not include overshooting convection. If the authors are aware of whether (or how) global models treat overshooting convection, it would be helpful to provide some discussion here.

4. As shown by previous cloud-resolving model studies (e.g. *Dauhut et al.*, 2018) the magnitude of irreversible hydration in the lower stratosphere increases as the maximum heights of overshoots extending into the stratosphere increase. It would be helpful for the authors to discuss this issue within the context of the current simulations. In addition, it would be useful to see how the distribution of overshoot maximum heights depends on the model spatial resolution.

**Review of "Simulation of convective moistening of extratropical lower stratosphere using a numerical weather prediction model" by Z. Qu et al.**

This manuscript uses cloud-resolving model simulations with different horizontal resolutions and different treatments of deep convection to investigate the physical processes leading to hydration of the lower stratosphere by overshooting deep convection. The manuscript is generally clear and well written. The results help clarify the roles of advection, wave breaking, and sublimation in hydration of the lower stratosphere. I have a number of minor comments I would like the authors to consider before publication.

**Suggestions for authors**

1. It would be helpful if the authors would provide some information about the simulated microphysical properties of the convective tops extending into the stratosphere. Specifically, quantitative information about the simulated ice concentrations and size distributions would be helpful. I realize the simulations use bulk parameterizations, but the two-moment scheme should provide ice concentrations and some measure of the width of the assumed size distribution. Realistic treatment of ice microphysics is important because the simulated convective hydration depends in part on the ice crystal size-dependent sedimentation and sublimation of ice in the lower stratosphere.

2. Page 1, line 30: The authors cite Anderson et al. (2012) here for the influence of water vapor on stratospheric chemistry. I believe earlier references such as *Solomon et al.* (1986) would be more appropriate.

3. Page 2, lines 26–20: I would recommend citing *Smith et al.* (2017) here.

3. Page 3, lines 17–19: It is my understanding that global models generally do not include overshooting convection. If the authors are aware of whether (or how) global models treat overshooting convection, it would be helpful to provide some discussion here.

4. As shown by previous cloud-resolving model studies (e.g. *Dauhut et al.*, 2018) the magnitude of irreversible hydration in the lower stratosphere increases as the maximum

heights of overshoots extending into the stratosphere increase. It would be helpful for the authors to discuss this issue within the context of the current simulations. In addition, it would be useful to see how the distribution of overshoot maximum heights depends on the model spatial resolution.

5. Page 4, line 29: It is confusing (and possibly misleading) to refer to the aircraft flight paths as "trajectories". It would be better to use terminology such as "flight path" or "aircraft altitude profile".

6. Page 5, lines 1–10: *Smith et al.* (2017) also did a trajectory analysis to determine the convective systems responsible for the observed lower-stratospheric water vapor plumes. It would be helpful if the authors compared the results/conclusions from the trajectory analysis done here with *Smith et al.* (2017).

7. Figure 4 is presumably a longitude slice through Domain A. An x-axis should be provided. Also, does this slice correspond to a particular latitude, or are the authors averaging over latitude within Domain A?

8. Page 8, line 24: The authors discuss the simulated moisture enhancement in the lower stratosphere. How is this enhancement calculated? Is a difference taken between the post-convection moisture field and the pre-convection field?

9. Page 9, lines 5–10: The authors discuss errors, uncertainties, and biases in the MLS $H_2O$ retrieval. However, my understanding is that the 100-hPa retrievals that are most relevant for this paper are in good agreement with observations.

10. Section 3.4: I believe this section could be better organized. There seems to be a fair amount of repetition, and the discussion seemed to meander. Perhaps this section could be more concise, and a sentence or two at the beginning outlining the analysis techniques would be helpful.

**References**

Dauhut, T., J.-P. Chaboureau, P. H. Haynes, and T. P. Lane (2018), The mechanisms leading to a stratospheric hydration by overshooting convection, *J. Atmos. Sci.*, *75*, 4383–4398.

Smith, J. B., D. M. Wilmouth, K. M. Bedka, K. P. Bowman, C. R. Homeyer, J. A. Dykema, M. R. Sargent, C. E. Clapp, S. S. Leroy, D. S. Sayres, J. M. Dean-Day, T. P. Bui, and J. G. Anderson (2017), A case study of convectively sourced water vapor observed in the overworld stratosphere over the united states, *J. Geophys. Res.*, *122*, doi:10.1002/2017JD026,831.

Solomon, S., R. R. Garcia, F. S. Rowland, and D. J. Wuebbles (1986), On the depletion of Antarctic ozone, *Nature*, *321*, 755–758.

———————————————————

---

## Author Comment (AC1) · 18 Dec 2019

Page 1, line 4: Maybe add here Riese et al. 2012. They showed also how small change in water vapor due to mixing processes change the radiative budget of the UTLS.

Reference added.

Page 2, line 24: Isentropic transport of water vapor due to planetary wave activity is also an important transport mechanism for transporting tropical tropospheric air into the lower extra-tropical stratosphere (see e.g. McIntyre and Palmer, 1983; Waugh, 1996; Homeyer and Bowman, 2012). I recommend to include this mechanism also in

the manuscript to complete all transport pathways.

Thanks! It's added into the manuscript.

Page 2, line 33: I recommend to add the paper of Lee et al., 2019. They performed also high-resolution model simulation of an overshooting event in the Asian monsoon region and showed how the moistening occurred and how the hydrated air was transported in the lower stratosphere.

Reference added.

Page 4, line 34: In summer times the standard value in the extra-tropical lower stratosphere is more 5 ppmv (see Zahn et al. 2014 Figure 5). I would recommend to change the text from 4 to 5 ppmv.

Done.

Page 6, line 21: Here you state the time of Fig. 4 to be at 19:49. In the figure caption it is stated 19:46. Please correct one of these times.

Correction is done. The correct time is 19:46 UTC.

Where there any cloud instrumentation aboard the ER-2 for measuring cloud number concentration or IWC? If yes, did you check if there were still ice crystals present at flight altitude in the domain B. That would be interesting to see, because than the ice crystals would have been transported over a longer distance in the stratosphere. This transport is shown by the Lee et al. 2019 and it would be interesting to see, if it occurred also in your case.

Good point! We have checked the ice water content for the ER-2 flight within domain B. There are some areas we found the presence of ice with the IWC between $1\times10^{-6}$ and $1\times10^{-3}$ g/m3 at lower altitudes between 14 and 15.5 km below the tropopause level. Fig. 1b shows the ice water content observed by the aircraft. The horizontal locations of these ice are marked by red dots in Fig. 2 on the aircraft path (gray line). We traced

the air parcels in the northern part of domain B from the altitude of 14 km back to 04:00 UTC when a more recent convection is happening in domain A. Particularly, the ice seen in the northeast part of domain B can be linked to the convection event as highlighted by the red trajectory. The back-tracing properties of this parcel are shown in Fig. 3. We can find that the formation of ice is near 04:00 UTC due to the convection (Fig. 3e). During the next 8 hours, the air parcel are slightly super-saturated (Fig. 3d). Therefore, the loss of ice is not due to the sublimation but the falling of ice to lower altitude. Near the time of 12:40 UTC, the relative humidity with regard to ice falls to below 1. All the remaining ice is quickly sublimated within about half an hour. Nevertheless, the ice is then reforming near 13:40 UTC due to the slow ascent of air with decreasing temperature and increasing relative humidity. The ice shown in Fig. 2 for the highlighted parcel is therefore not the ice originally formed during the convection, but formed later during the ascent of air about 6 hours before. We can also observe the impact of the dehydration due to the formation and the falling of ice near 15:40 UTC with a decrease of water vapor mixing ratio of $\sim$20 ppmv.

Similarly, in the GEM simulation even in a higher altitude above the tropopause we still found this formation of ice although with smaller ice water content (in the order of $\sim$ 1x10-5, see updated Fig. 8 in the manuscript). The origin of the humid air parcel is often linked to the convection that happened before. In the case of Fig. 8 (manuscript), it is the convection started from 25 Aug in the simulation. Since the formation of ice in higher altitudes above tropopause is much less significant than in lower altitudes. Contrary to the dehydration effect in Fig. 3 near 15:40 UTC, we didn't find a significant impact of dehydration in high altitudes (e.g. Fig. 8 of the manuscript). Therefore, this will not change our conclusion by linking the water vapor injected in domain A with the simulated/observed water vapor in domain B for the altitudes above the tropopause.

Some new texts are added into the manuscript, the Fig 2 and 3 here will be added to the SI materials:

"One particular note for Fig. 8 is the formation of ice shortly after 19:40 UTC 27 Aug

when the humid air parcel slowly ascends with decreasing temperature and increasing relative humidity with regard to ice. At these altitudes, the ice water content is relatively low ($\sim$1x10-5 g m-3). The ice particles will gradually fall to a lower altitude and eventually be sublimated again. This process will partly dehydrate the upper layer of the atmosphere where the ice is forming, and later hydrate the lower atmospheric layer through ice sublimation. However, this dehydration has minor impact for the air parcels above the tropopause. We observe that the water vapor mixing ratio of the air parcel at the 15.5 km altitude (i.e., the upper layer) increased slightly after 19:40 UTC (Fig. 8a). This might be the results of the mixing with the adjacent air in the northeast side of the parcel which is more humid (pointed by the black arrow in Fig. 7). Due to the limited impact of dehydration through ice formation above the tropopause, our interpretation of linking the stratospheric water vapor injected in domain A and water vapor simulated/observed in domain B is therefore not affected in a significant way by the ice formation process."

"For the atmospheric layer under the tropopause between the altitude of 13.5 and 14.5, the horizontal wind speed increases significantly. The back-tracking results show that the humidity and ice field in the northern part of Domain B are linked to the convection initiated at the beginning of 27 Aug in domain A. The locations of ice water content in domain B from the simulation partly agree with what are observed by the aircraft in Fig. 2b. Based on the back tracing results, we noticed that the ice in Domain B is not originally formed during the convection, but later during the slow ascent of the humid air parcel. This is similar to the hydration/dehydration process discussed above but at a lower altitude below the tropopause. The ice formed at this lower altitude is more abundant (in the order of 1x10-3 g m-3). The impact of dehydration (ice formation and falling) at this level is significant which can be seen in Fig. SI.3 near 15:40 UTC with an amplitude of about 20 ppmv. The readers are referred to the supplementary materials for more discussions on this topic (Fig. SI.2, SI.3)."

Page 9, lines 5-10: Maybe it is worth a mention that the averaging kernels of

limb sounders like MLS smear out the strong vertical gradient in water vapor at the tropopause (Hegglin et al.,2013).

Added to the manuscript.

Page 9: The comparison between GEM and MLS is not really done in a balanced way. The difference is partly larger (> 100%) than the biases which are reported in the literature. These strong differences can be hardly explained by just mentioned the possible bias of MLS water vapor. It could also be a result of the model simulation. For example, warmer temperatures in comparison to MLS could lead to slower ice crystal growth and thus less dehydration and thus higher gas-phase water, which could be transported into the lower stratosphere.

Thanks, this is a very good point! We modified the text as follow:

"We applied the averaging kernel of MLS on the mean profiles of GEM simulated humidity and temperature within the 100x100 km regions centered on the MLS footprints. The comparison here suggests that both model simulations give higher estimations of water vapor content in the UTLS comparing to MLS retrievals, although the higher-resolution simulation better approximates the satellite observations. It is also found that GEM slightly overestimated the temperature comparing to MLS retrievals. This suggests that warmer temperatures in comparison to MLS could lead to slower ice crystal growth and thus less dehydration and thus higher gas-phase water. The spatial-temporal errors of the model simulation, e.g. shifted convection area, and time, etc., might also contribute to the discrepancies between the GEM and MLS profiles. Further, the lower value of water vapor content from MLS near the level of 160 hPa may be subject to the aforementioned negative bias in the MLS data."

The averaging kernels of MLS are applied to the GEM profiles for a more suitable comparison. The Fig. 6c and 6d of the manuscript are modified accordingly.

Can you please comment on the following questions and suggestions? Which Version

of MLS data are you using (this would be also nice to mention in the text)?

The version used in this study is v4.2. We added this information to the manuscript.

Did you apply the averaging kernels of MLS onto the GEM profile? Otherwise a fair comparison is barely possible. Why not using the exact location of MLS and applying the averaging kernels?

The averaging kernel was not applied in the original version. We added the comparison applying the averaging kernel in the updated version (see answers above). This does not change the comparison results or any conclusion.

Why are the profiles of water vapor (panel a/c/e) and also temperature (panel b/d/f) so different? They should both represent air masses moving from domain a to domain b as shown by the trajectories. It seems that the situation is strongly variable. Can please discuss in the text about the standard deviation of the mean profiles to get a better feeling on the variability.

One factor causing the differences in profiles is that there is a slow but persistent vertical movement of air as shown both in Fig. 8c in the manuscript and Fig. 3c in this response document. This will change the form of the profiles. For example, the 'bump' in humidity profile in Fig. 6a (manuscript) is at the level of 16 km, while it moves to 16.7 km in Fig. 6e (manuscript) which is coherent with the slow ascending motion. Other factors such as the large circulation, radiative heating/cooling and the significant differences in latitude which results a warmer tropopause temperature in the north (domain A) and a cooler one in the south (domain B) (Fig. 4).

For better understanding of the deviation between GEM and MLS, I would also recommend to add the location of the MLS profile also into the map in Figure 7.

Done

Figure 8: You show some parameters of an individual trajectory in this figure. The water vapor amount with ~20 ppmv is quite high and the temperatures are cold between

203-206K, which could create conditions with supersaturation wrt ice. and therefore, additional ice formation. Do you see any signs of ice formation in the lower stratosphere along these trajectories? This could be important, because it could partly dehydrate the previously hydrated air masses.

For this particular air parcel, the formation of very thin ice does happen during the ending hours. More discussion is added into the manuscript for the dehydration effect by the formation of the ice (see the answer above). The ice water content and relative humidity are added to the Fig. 8 in the manuscript.

Page 10-11: For me it is not clear, which part of the equation 5 account for ice sublimation/transport? Because in the equation only q, which stands for water vapor, is considered. Can you please better explain how you estimated the change due to advection and ice sublimation as stated in lines 16-20.

We clarify that the equation is used for analyzing and understanding the direct transport. In the manuscript, we try to use the Reynolds decomposition to investigate the importance of injection due to gravity wave breaking with regard to the direct transport only. We argue that for the high-resolution simulation, the majority of the direct transport is linked to the gravity wave breaking. Ice sublimation is therefore not taken into account in Eq. 5.

Page 12-14: Where does the sublimation occurs in the high-resolution models? Is it directly in the overshoot or are the ice crystals first mixed into the lower stratosphere by wave breaking and small-scale mixing and then sublimate? Which brings me to a further question, if ice crystals are transported along the trajectories in the lower stratosphere? Perhaps, you can add this information also into the manuscript.

In the high-resolution simulations, ice was firstly brought to the lower stratosphere within the cloud overshooting tops. There the ice cannot be sublimated efficiently. During the fall of the overshooting top the gravity wave breaking happens, a fraction of ice in the overshooting top will be brought into lower stratosphere in an irreversible way (no

downward movement to bring then back to troposphere, only relying on sedimentation). Further, the humid air transported into the lower stratosphere might be supersaturated depending on the temperature of the reached altitude. This will be a second contribution to the ice plume. These plumes will gradually be sublimated through mixing with dryer air or be sediment to a lower altitude. In our simulation at 1.0 km grid-spacing, these plumes will not last for a long time (generally less than 1 hour) before they disappeared completely near the convective zone. We added the description for ice staying in lower stratosphere in the manuscript:

"Ice plumes are also formed near the areas where the gravity wave breaking happens. Two sources are found: the direction transport of ice and the formation of ice under supersaturation condition within humid plumes. The sizes of ice plumes are generally smaller than those of the water vapor plumes, because ice will be completely sublimated/sediment within a short period of time, generally within one hour."

Page 13/14: What does a negative sublimation tendency mean? I would guess it is additional ice formation or particle growth. Or is it both?

The notion 'sublimation' presented in the manuscript denotes the dynamical effect of both ice sublimation and vapor deposition. The negative value signifies that the vapor deposition is faster than the ice sublimation, hence dehydration. We realize that the use of 'sublimation' in might cause confusion. Additional explanation is added to clarity the definition of sublimation:

"Ice sublimation (hydration) and vapor deposition on ice (dehydration) are two opposing microphysical processes competing for dynamical balance. We use hereinafter 'sublimation' to denote the combined effect of these two processes. The positive value signifies that the ice sublimation is faster than the vapor deposition, and the negative value signifies the other way around."

Page 14: I agree with your conclusion that ice sublimation occurs less pronounced in the high-resolution model because of existence of ice crystals mostly in the overshoot-

ing core. In the relative humidity distribution (Figure 12 b) I would expect higher fraction of IWC in the sub-saturated region (Rhi < 1) for the low resolution compared to the high resolution, if there is a much higher sublimation rate. Do you have an explanation for the agreement of IWC distribution in sub-saturation for all three model setups?

In 10 km simulation (Fig. 11d and 11g of the manuscript), we can find that the majority of the ice at this altitude is located in an area of slight supersaturation (rhi ~1.09). This means that the majority of the ice is not sublimating, but slowly growing by vapor deposition. This can be seen in Fig.11a (manuscript) that in the cloudy area the vapor deposition is happening although with slow rates (light blue area). The areas where we find high ice sublimation rates are near the northern edges of the cloudy area due to the mixing with the dry stratospheric air. These areas have an important contribution for the mean sublimation rate in the domain A, and their relative humidity tends to be sub-saturated, although their ice mass fraction in domain A is not significant. This can be one of the reasons why in Fig.12b (manuscript) the majority of the ice is slightly supersaturated rather than sub-saturated.

Another important factor is the way GEM calculates the dynamic and physical processes. At a given time step, GEM calculates the dynamic processes first (advection, etc.) and then the physical processes (microphysics, etc.). After the dynamic calculations, the sub-saturation should be very pronounced on the edge of the cloudy area, but the properties are not saved until the physical processes are finished which reduces quickly the sub-saturation due to the sublimation. Considering that in the 10 km simulation the time step is set to 5 min which is long enough to reduce the sub-saturation produced by the mixing with dry stratospheric air (dynamics).

Technical comments/suggestions: Page 8, line 8: Citation should be Vömel et al.

Done

Page 14, line 9: It is more common to use the term supersaturated instead of oversaturated.

Done

Figure 2: Can you please include the date of the ER-2 measurements into the figure caption.

Done

Figure 4: The comprehensibility of the vertical wind speed in panel d would be better, if you choose a color scale centered with the color white at the value of 0 and with positive/negative values in two different colors (e.g. red and blue).

Done

Figure 5: Same suggestion above for panel c and d.

Done

Figure 12, caption: "relative humidity" instead of "relatively humidity"

Done

**(a) Water vapor mixing ratio**

**(b) Ice water content**

**(c) Altitude**

**(d) Temperature**

**Fig. 1.** ER-2 aircraft observation of a) water vapor mixing ratio, b) ice water content, c) altitude and d) air temperature in K, on 27 Aug 2013. Green dots: ice observed.

**GEM 2.5 km Back-Trajectories, IWC (g m⁻³)**

$$\text{GEM 2.5 km Back-Trajectories, IWC (g m}^{-3})$$

**Fig. 2.** Background: ice water content (∼14 km, 1940 UTC 27 Aug); gray line: ER-2 aircraft path (red dots: ice); circles: starting points at 19:40 UTC; triangles: ending point at 04:00 UTC.

[Figure]

**Fig. 3.** The changes of the properties of the highlighted air parcel in Fig. R2 along its back trajectory from 19:40 UTC to 03:00 UTC 27 Aug.

**TT(K), 2.5 km, Time: 2880m**

**Fig. 4.** The temperature field at the altitude of 16 km, at 23:00 UTC 25 Aug. Magenta solid line: domain A (profile a/b); magenta dash-dot line: domain B (e/f); red diamonds: MLS footprints (c/d).

---

## Author Comment (AC2) · 18 Dec 2019

1. It would be helpful if the authors would provide some information about the simulated microphysical properties of the convective tops extending into the stratosphere. Specifically, quantitative information about the simulated ice concentrations and size distributions would be helpful. I realize the simulations use bulk parameterizations, but the two-moment scheme should provide ice concentrations and some measure of the width of the assumed size distribution. Realistic treatment of ice microphysics is important because the simulated convective hydration depends in part on the ice crystal size-dependent sedimentation and sublimation of ice in the lower stratosphere.

[Figure]

The microphysics scheme used here predicts the ice water content (IWC) and the number concentration for 4 solid hydrometeors: ice, snow, graupel and hail. The particle size of solid categories is assumed to be gamma distribution. For the detailed parameters used in the distribution please refer to Milbrandt et al. (2005a, 2005b). Here, in order to present the results in a succinct way, we calculated the effective radius (the ratio of the third to the second moment of a droplet size distribution) for each solid category predicted by the 1.0 km simulation and weighted them with the mass. The 2D histogram of this mass weighted effective radius is shown in Fig. 1 for 19:46 UTC, 25 Aug (the same time as for Fig. 4 in the manuscript), at the altitude of ∼15.5 km in domain A. At the other altitudes between 15 and 16, we find similar distributions (not shown).

The cloud overshooting tops often contain high ice water content, e.g. IWC>0.5 g m-3. In Fig. 1, we find that these air parcels containing high IWC are very few (light blue region circled). The mass weighted effective radius is between ∼300 to ∼700 microns which suggests that there are large particles and they will fall faster. On the other hand, for the thin ice clouds, e.g. the ice plumes, with IWC<0.1 g m-3, they occupy a much larger area comparing to the overshooting tops as they are more frequently encountered (red color). The effective radius of these solid particles is small, mostly less than 30 microns as shown in Fig. 1.

For our current study, we don't have direct observation data of ice number concentration and ice water content during the convection time. Therefore, it will be difficult to validate the results of GEM. We considered this an interesting subject for a further study, e.g. to use other aircraft in situ observations for a different case for which we have good measurements near the convection. We add in the manuscript these descriptions:

"The cloud ice properties are different in the overshooting tops and in the thin ice plumes. At the altitude of ∼15.5 km (∼ 1 km above the tropopause) within domain A at 19:42 UTC 25 Aug, the ice water content in the overshooting tops is relatively

high with values from ~0.5 g m-3 up to ~2.8 g m-3. In the thin ice plumes the ice water content is generally lower than 0.1 g m-3. We calculated the effective radius for each solid category, e.g. ice crystals, rain, snow, graupel and hail. We find that in the cloud overshooting tops, the mass weighted effective radius for ice increases with the ice water content from ~300 to ~700 microns. On the other hand, the mass weighted effective radius for thin ice plume is usually lower than 30 microns. The area of cloud overshooting top occupies only 2.3% of the cloudy area but contains 68% of the total ice mass at this altitude."

2. Page 1, line 30: The authors cite Anderson et al. (2012) here for the influence of water vapor on stratospheric chemistry. I believe earlier references such as Solomon et al. (1986) would be more appropriate.

Done.

3. Page 2, lines 26–30: I would recommend citing Smith et al. (2017) here.

Done.

4. Page 3, lines 17–19: It is my understanding that global models generally do not include overshooting convection. If the authors are aware of whether (or how) global models treat overshooting convection, it would be helpful to provide some discussion here.

Many GCM and global NWP models employ the mass flux approach to represent deep convection. In this approach, the updraft characteristics are calculated using a steady state plume model. This includes solving an equation for the updraft vertical velocity as a function of the evolving buoyancy of the entraining plume. The cloud top will be defined as where the vertical velocity approaches zero. This level is always above the level of neutral buoyancy. Therefore, these schemes do represent in a very simplified manner overshooting convection. How high this convection reaches depends on the environmental characteristics as well as on many parameters of the convection

scheme, namely on how entrainement and detrainment is calculated. Other complex phenomena near the tropopause during the convection are not parameterized, e.g. the falling of cloud overshooting tops, gravity wave breaking and formation of jumping cirrus, as well as the trapping of ice within the supersaturated cloud overshooting tops which inhibits ice sublimation. We will discuss all these points in details in the later parts of the manuscript. They warrant also further investigations and eventually improvements of the way of parameterizing deep convection near the tropopause. We add in this paragraph a brief discussion:

"In the global NWP and GCM models, the deep convection is parameterized using mass flux approach. The complex phenomena near the tropopause during the convection are parameterized in a simplified manner, e.g. overshooting convection, or not parameterized, e.g. the falling of cloud overshooting tops (not sedimentation), gravity wave breaking and formation of jumping cirrus, etc."

5. As shown by previous cloud-resolving model studies (e.g. Dauhut et al., 2018) the magnitude of irreversible hydration in the lower stratosphere increases as the maximum heights of overshoots extending into the stratosphere increase. It would be helpful for the authors to discuss this issue within the context of the current simulations. In addition, it would be useful to see how the distribution of overshoot maximum heights depends on the model spatial resolution.

For the fully solved high resolution simulations, the maximum height of the cloud overshooting top depends on the model horizontal resolution. In our simulated case, the maximum heights are 16.64 and 16.96 km for 2.5 and 1.0 km simulation. For 0.25 km simulation, the early stage of the convection was simulated in the limited green box in Fig. 1 with a maximum height of 16.64 km. The irregularity here is probably due to the lack of the latter period of the convection for the 0.25 km simulation (in the other simulations, the maximum heights are often found in the later period). Nevertheless, we can conclude that the higher the model horizontal resolution is, the higher the cloud overshooting top will reach in the lower stratosphere. As for the 10 km simulation with

parameterized convective cloud, as mentioned in the previous question, the height of cloud overshooting top depends on the environmental characteristics as well as on many parameters of the convection scheme, namely on how entrainement and detrainment is calculated. It will not be reasonable to compare directly the 10 km simulation with the other high resolution simulations, even though the maximal cloud top height of 10 km simulation (16.13 km) is lower than all the other high resolution simulations. We modified the manuscript and discussed this point later in the manuscript:

"It is not a surprise that the higher resolution NWP models tend to produce stronger direct vertical transports across the tropopause because, as shown in Subsection 3.1, the transport is closely related to the strength of overshooting and the breaking of the gravity waves. Similar to what was found by Weisman et al. (1997), we find in our GEM simulations that the simulated maximal vertical wind speed is inversely proportional to the horizontal grid-spacing of the NWP model. The stronger vertical wind speed in the convection updraft leads to higher cloud overshooting top. In our cases with high resolution simulation, the maximum cloud top altitude is 16.64 and 16.96 km for 2.5 and 1.0 km simulation respectively. We find that the stronger overshooting wind speed in the higher resolution simulations leads to favorable conditions for gravity wave breaking (see the discussions in Subsection 3.1) and thus more direct vertical transport. This agrees with Dauhut et al. (2018). In total, the direct vertical transport of water vapor contributes to 40% of the total transport at the tropopause level for the 2.5 km simulation and makes up to 89% for the 1.0 km simulation."

6. Page 4, line 29: It is confusing (and possibly misleading) to refer to the aircraft flight paths as "trajectories". It would be better to use terminology such as "flight path" or "aircraft altitude profile".

Done.

7. Page 5, lines 1–10: Smith et al. (2017) also did a trajectory analysis to determine the convective systems responsible for the observed lower-stratospheric water vapor

plumes. It would be helpful if the authors compared the results/conclusions from the trajectory analysis done here with Smith et al. (2017).

Smith et al. (2017) focused on the high water vapor content observed by the ER-2 aircraft between ∼19:30 to ∼19:50 UTC (the third dive in Fig. 2 of the manuscript) in the north-most part of domain B at the level near ∼100 hPa. The source of the humid air parcels observed there are mostly traced back to the convection east to the Lake Superior between 26 Aug 2013 21:00 UTC and 27 Aug 2013 12:00 UTC (hereinafter named convection Day2). Our back trajectory calculation gives similar results as shown for a humid air parcel from the northeast of domain B (Fig. 2 in this response, dashed back tracking line) which is traced back to the area of convection Day2. We note that the simulated convection initiated at the end of 26 Aug is slightly north to the real convection observed by satellite image (a location error in the simulated convection). This justifies the use of average of domain B, as opposed to individual location(s), for the evaluation so that humid areas linked to the convection Day2 are always included.

One of the foci of current paper is the average of water vapor content profile in the domain B. As explained above, this is designed to tolerate the spatial-temporal errors of the model simulations. For the southwest of domain B, the humid air parcels are mostly traced back to the convection that happened east to the Lake Superior between 25 Aug 17:40 UTC and 26 Aug 05:40 UTC (hereinafter named convection Day1). This is shown in Fig. 2 by the solid red back trajectory line. Having inspected these trajectory results, we found that the contribution for the average water vapor content in domain B by the convection Day1 (southwest) is more important than that of the convection Day2 (northeast). We therefore focused more on the convection Day1 in most parts of the manuscript, e.g. the convection shown in Fig. 3 of the manuscript as well as the budget analysis in section 3.4.

We added in the manuscript a brief discussion for the comparison with the results of Smith et al. (2017):

"Using this technique, we find that the large water vapor anomalies observed by the aircraft in Domain B on 27 Aug 2013 originated from two deep convection events. The first one began at the end of 25 Aug and ended at the beginning of 26 Aug in Domain A (100ËŽ W to 87.5ËŽ W, 46ËŽ N to 50ËŽ N, ∼860x445 km2) as illustrated in Fig. 1 (see more discussions below). This convection has major contribution to the water vapor content in the lower stratosphere of domain B. The second source is the convection began at the end of 26 Aug and ended at the beginning of 27 Aug in Domain A. This second convection increases the water vapor content in the northern part of domain A. This agrees with the results of Smith et al. (2017) in which the humid air parcels observed by the aircraft near 19:40 UTC (Fig. 1, northeast of domain B) are traced back to the convection began at the end of 26 Aug."

8. Figure 4 is presumably a longitude slice through Domain A. An x-axis should be provided. Also, does this slice correspond to a particular latitude, or are the authors averaging over latitude within Domain A?

The cross-section presented in Figure 4 of the manuscript is not along a longitude but along a skew line. We added a separated panel (e) in Figure 4 of the manuscript to show the location of this cross-section within the domain A.

9. Page 8, line 24: The authors discuss the simulated moisture enhancement in the lower stratosphere. How is this enhancement calculated? Is a difference taken between the post-convection moisture field and the pre-convection field?

Thanks, it's a good question. The profiles shown in Fig. 6a in the manuscript is for the average water vapor mixing ration during the 5 hours' evaluation time within domain A. The horizontal transport of water vapor should therefore have an impact on those profiles. In a more direct way, by comparing the averaged profiles before and after the convection in Fig. 3, we can see an enhancement of water vapor mixing ratio above the altitude of 16 km, but not between 15 and 16 km for the two high resolution simulations. This is mostly due to the horizontal transport of water vapor out of domain A through the

western and southern borders. For the altitude between 15 and 16 km, the horizontal wind speed is fairly high (Fig. 5e of the manuscript). Therefore, the impact on the comparison of vertical profile is significant. On Fig. 4a, the water vapor mixing ratio field on the level of ∼15.5 km is shown. At this time the convection just began. We find humid air in majority part of the domain A, except the westmost part. These humid areas are linked to the convection happened during the previous day (24 Aug). During the evaluation period on 25 Aug, the convection we focused on injected a large amount of water vapor into this altitude. Meanwhile, most of the humid air presented at the beginning of the evaluation time is moving out of the domain. This caused eventually a decrease of the average humidity within domain A as shown in Fig. 3a.

After rethinking about the statement here about the enhancement of water vapor, we considered that it is not suitable to mention it here because the conclusion could not be derived directly form Fig. 6a of the manuscript. A better way to evaluate the enhancement of water vapor is to calculate how much water vapor is transported vertically through the tropopause. This calculation is discussed in detail in the later section of the manuscript (section 3.4, budget analysis).

The main purpose of Fig. 6a (manuscript) is to show that 10 km simulation produce clearly moister lower stratosphere than the two high resolution simulations. We modified the text as follow:

"We further examine the water vapor fields simulated by GEM at different horizontal grid-spacings. Fig. 6a and 6b show the mean vertical profiles of water vapor volume mixing ratio and temperature within the afore-defined Domain A and 5-hour time window. All the simulations show irregular moisture profiles near 16 km, where the vertical trend of the humidity profiles bends and produces 'bumps' (elevated water vapor contents) above the tropopause (indicated by the circles in Fig. 6, hereinafter the tropopause is defined by the altitude where the mean lapse rate $\Gamma$ within Domain A and 5-hour time window decreased to 2ĚŽC km-1 or less)."

10. Page 9, lines 5–10: The authors discuss errors, uncertainties, and biases in the MLS H2O retrieval. However, my understanding is that the 100-hPa retrievals that are most relevant for this paper are in good agreement with observations.

Agreed. In response to the comments of the other reviewer, we updated the way of comparison between MLS data and GEM data by applying the averaging kernel of MLS to GEM profiles. This allows a more coherent comparison as suggested by another referee. The results are shown in the updated Fig. 6c, 6d of the manuscript (or the Fig. 5c and 5d of the response). For the pressure levels near 100 hPa, we observed moister air from both GEM simulations which might indicate the overestimation of water vapor from the model simulation. The 10 km simulation gives even higher water vapor mixing ratio than the 2.5 km simulation and MLS, which evidences the positive bias in the low resolution model. On the other hand, near the levels of 160 hPa, we can see significant differences between the model simulations and the MLS retrievals. At these lower levels, the MLS data might subject to the negative bias reported by Hegglin et al. (2013), Vömel et al. (2007) and Livesey et al. (2018). This makes the comparison more uncertain; hence the validation against aircraft data is used.

We modified the text in the manuscript concerning the comparison between model simulations and MLS data:

"Figure 6c, 6d show the comparisons between the GEM simulations after applying averaging kernels of MLS and MLS retrievals (v4.2). Because of the scarcity of the collocated satellite data and also the afore-mentioned mismatch in time and location of the simulated convective system, we conduct the comparison with respect to area-averages rather than individual samples. The MLS measurements used here include five MLS footprints located between [38 N, 45 N] and [95 W 93 W], taken on 26 Aug 2013 around 19:00 UTC, about 15 hours after the dissipation of the convection system (Fig. 7, red diamonds). We applied the averaging kernel of MLS on the mean profiles of GEM simulated humidity and temperature within the 100x100 km regions centered on the MLS footprints. The comparison here suggests that both model simulations give

higher estimations of water vapor content in the UTLS comparing to MLS retrievals, although the higher-resolution simulation better approximates the satellite observations. It is also found that GEM slightly overestimated the temperature comparing to MLS retrievals. This suggests that warmer temperatures in comparison to MLS could lead to slower ice crystal growth and thus less dehydration and thus higher gas-phase water. The spatial-temporal errors of the model simulation, e.g. shifted convection location or time, might also contribute to the discrepancies between the GEM and MLS profiles. Furthermore, the lower value of water vapor content from MLS near the level of 160 hPa may be subject to the aforementioned negative bias in the MLS data."

11. Section 3.4: I believe this section could be better organized. There seems to be a fair amount of repetition, and the discussion seemed to meander. Perhaps this section could be more concise, and a sentence or two at the beginning outlining the analysis techniques would be helpful.

Some changes of the structure are made. Please refer to the modified manuscript for details.
* * *
[Figure]

**Fig. 1.** 2D histogram of cloud ice in domain A at the altitude of 15.5 km, 19:46 UTC 25 Aug. x-axis: ice water content; y-axis mass weighted effective radius. 50 bins are used in each axis.

**GEM 2.5 km Back-Trajectories, $q_v$ (ppmv)**

**Fig. 2.** Back tracking from 19:40 UTC 27 Aug to: 22:40 UTC, 25 Aug for convection Day1, and to 21:00 UTC 26 Aug for convection Day2.

[Figure]

**Fig. 3.** The mean profiles of water vapor volume mixing ratio (qv) and temperature (T) for Domain A before (solid lines for 18:00 UTC, 25 Aug) and after (dotted lines for 23:00 UTC, 25 Aug) the convection.

[Figure]

**Fig. 4.** The water vapor mixing ratio field at the altitude of $\sim$15.5 km before (a) and after (b) the convection on the day of 25 Aug. The magenta box represents the domain A.

[Figure]

**Fig. 5.** Same to Figure 6 in the manuscript, except c), d) mean profiles (qv and T) after applying averaging kernels of MLS on GEM 2.5 km, 10 km simulation.